# Temperature controls production but hydrology controls export of dissolved organic carbon at the catchment scale

Hang Wen[1], Julia Perdrial[2], Benjamin W. Abbott[3], Susana Bernal[4], Rémi Dupas[5], Sarah E. Godsey[6], Adrian Harpold[7], Donna Rizzo[8], Kristen Underwood[8], Thomas Adler[2], Gary Sterle[7], Li Li[1]*

[1]Department of Civil and Environmental Engineering, The Pennsylvania State University, University Park, PA 16802, USA

[2]Department of Geology, University of Vermont, Burlington, VT 05405, USA

[3]Department of Plant and Wildlife Sciences, Brigham Young University, Provo, UT 84602, USA

[4]Center of Advanced Studies of Blanes (CEAB-CSIC), Accés Cala St. Francesc 14, 17300, Blanes, Girona, Spain

[5]INRA, UMR1069 SAS, Rennes, France

[6]Department of Geosciences, Idaho State University, Pocatello, ID 83201, USA

[7]Department of Natural Resources and Environmental Science, University of Nevada, Reno, NV 89557, USA

[8]Department of Civil and Environmental Engineering, University of Vermont, Burlington, VT 05405, USA

*Correspondence to: Li Li (lili@engr.psu.edu)

**Abstract**: Lateral carbon flux through river networks is an important and poorly-understood component of the global carbon budget. This work investigates how temperature and hydrology control the production and export of dissolved organic carbon (DOC) in the Susquehanna Shale Hills Critical Zone Observatory in Pennsylvania, USA. Using field measurements of daily stream discharge, evapotranspiration, and stream DOC concentration, we calibrated the catchment-scale biogeochemical reactive transport model BioRT-Flux-PIHM, which met the satisfactory standard of the Nash-Sutcliffe efficiency (NSE) > 0.5. We used the calibrated model to estimate and compare the daily DOC production rates ($R_p$; the sum of local DOC production rates in individual grid cells) and export rate ($R_e$; the product of concentration and discharge at the stream outlet, or load).

Results showed that $R_p$ varied by less than an order of magnitude through time, primarily depending on seasonal temperature. In contrast, $R_e$ varied by more than three orders of magnitude, strongly associated with variation in discharge and hydrological connectivity. In summer, high temperatures and evapotranspiration dried and disconnected hillslopes from the stream, driving $R_p$ to its maximum but $R_e$ to its minimum. During this period,

the stream only exported DOC from the organic-poor groundwater and from organic-rich soil water in the swales bordering the stream. Produced DOC accumulated in hillslopes and was later flushed out during the wet period (winter and spring) when $R_e$ peaked as the stream reconnected to a greater uphill area, and $R_p$ reached its minimum.

The model reproduced the observed concentration-discharge (C-Q) relationship characterized by an unusual flushing-dilution pattern with maximum concentration at some intermediate discharge. Sensitivity analysis indicated that this nonlinearity was caused by shifts in relative importance of different source waters to the stream across flow conditions. At low discharge, stream water reflected the chemistry of organic-poor groundwater. At intermediate discharge, stream water was dominated by the organic-rich soil water from swales. At high discharge, the stream reflected uphill soil water with intermediate DOC concentration. This pattern persisted regardless of DOC production rate as long as the contribution of deeper groundwater flow remained low (<18% of the streamflow). When groundwater flow increased above 18%, the stream water mixed a comparable amount of groundwater and swale soil water such that the maximum DOC concentration at intermediate discharges did not show up. The C-Q patterns therefore switched to a flushing-only pattern with DOC concentration increasing with discharge. In hot and dry conditions, the catchment served as a producer and storage reservoir for DOC, transitioning into a DOC exporter in wet and cold conditions. This study illustrates how different controls of DOC production and export - temperature and hydrological flow path, respectively - can create temporal asynchrony at the catchment scale. Future warming and increasing hydrological extremes could accentuate this asynchrony, with DOC production occurring primarily during dry periods and lateral export of DOC dominated by a few major storm events.

## 1. Introduction

Soil organic carbon (SOC) is the largest terrestrial stock of organic carbon, containing approximately four times more carbon than the atmosphere (Stockmann et al., 2013; Hugelius et al., 2014). Understanding SOC balance requires consideration of lateral fluxes in water, including dissolved organic and inorganic carbon (DOC and DIC), and vertical fluxes of gases such as $CO_2$ and $CH_4$ (Chapin et al., 2006). Both lateral and vertical fluxes influence SOC mineralization to the atmosphere (Campeau et al., 2019), although lateral fluxes are arguably less understood and integrated into Earth system models (Aufdenkampe et al., 2011; Raymond et al., 2016). Lateral fluxes from terrestrial to aquatic ecosystems are similar in magnitude to net vertical fluxes (Zarnetske et al., 2018; Regnier et al., 2013; Battin et al., 2009), highlighting the importance of quantifying the controls of lateral carbon (C) flux. In addition to its role in the global C cycle, DOC is an important water quality parameter that may mobilize metals and contaminants as well as impose challenges for water treatment when DOC is abundant (Sadiq

and Rodriguez, 2004; Bolan et al., 2011). Finally, DOC regulates food web structures by acting as an energy source for heterotrophic organisms and interacts with other biogeochemical cycles (Malone et al., 2018; Abbott et al., 2016a).

SOC decomposition and DOC production have been studied extensively (Abbott et al., 2015; Bernal et al., 2002; Hale et al., 2015; Humbert et al., 2015; Lambert et al., 2013; Neff and Asner, 2001), yet the interactions between SOC and DOC and their response to climate change at catchments or larger scales remain unresolved (Kicklighter et al., 2013; Abbott et al., 2016b; Laudon et al., 2012; Clark et al., 2010; Evans et al., 2005). Some regions have experienced long-term increases in DOC, potentially due to recovery from acid rain or climate-

induced changes in temperature ($T$) and hydrological flow (Laudon et al., 2012; Perdrial et al., 2014; Evans et al., 2012; Monteith et al., 2007). Others have observed decreases or no changes (Skjelkvale et al., 2005; Worrall et al., 2018). Generally, the linkages among SOC processing, hydrological conditions, and DOC export or concentration remain poorly understood. Recent analyses indicate that the relationship between DOC concentration and discharge (C-Q) at stream outlets is primarily positive (Moatar et al., 2017; Zarnetske et al.,

2018). Approximately 80% of watersheds in the U.S. and France show a flushing C-Q pattern (i.e. stream DOC concentration increases with discharge) whereas the rest show dilution (decreasing DOC with discharge) or chemostatic behavior (little concentration change with discharge). These C-Q patterns generally correlate with catchment characteristics, including topography, wetland area, and climate characteristics (Moatar et al., 2017; Zarnetske et al., 2018), but it remains uncertain how hydrological and biogeochemical processes regulate SOC

decomposition, DOC production, and DOC export (Jennings et al., 2010; Worrall et al., 2018). This gap in process understanding limits the integration of lateral carbon dynamics into projections of future ecosystem response to change.

Stream DOC can be influenced by a variety of factors that control SOC decomposition and DOC production rates in soils. DOC production generally increases as $T$ increases; but there may be multiple thermal

optima and the local rates can vary with SOC characteristics, soil type, and soil biota (Davidson and Janssens, 2006; Jarvis and Linder, 2000; Yan et al., 2018; Zarnetske et al., 2018). DOC production rates can exhibit low temperature sensitivity in highly weathered soils with high clay content (Davidson and Janssens, 2006). They have also shown to increase with soil water content in sandy-loam soils (Yuste et al., 2007) and to have an optimum with volumetric water content ~0.75 in fine sands (Skopp et al., 1990). Because DOC export at the

catchment scale is the product of discharge and DOC concentration, it may differ from local DOC production rates in complex ways. For example, high $T$ can produce peak soil water DOC concentration but not necessarily stream concentration or export, due to temporal or spatial mismatches (D'Amore et al., 2015). These confounding

factors present significant challenges to quantifying the predominant mechanisms that regulate DOC production and export under varying environmental conditions.

One flexible approach to understanding DOC production and export is the use of reactive transport modeling (RTM). These models integrate multiple production, consumption, and export processes, potentially allowing quantification of individual and coupled processes (Steefel et al., 2015; Li, 2019). The use of RTMs complements statistical regression tools for identification of factors influencing DOC dynamics (Correll et al., 2001; Herndon et al., 2015; Zarnetske et al., 2018). Historically, RTMs have been used in groundwater systems, where direct observations are particularly challenging (Kolbe et al., 2019; Li et al., 2009; Wen and Li, 2018; Wen et al., 2018). At the catchment scale, biogeochemical modules have been developed as add-ons to hydrological models. For example, a DOC production module was coupled to the HBV hydrological model, using a static SOC pool that emphasized the influence of active-layer dynamics and slope aspect (Lessels et al., 2015). The INCA-C (Futter et al., 2007) and extended LPJ-GUESS (Tang et al., 2018) models have investigated the importance of land cover in determining DOC terrestrial routing and lateral transport. Terrestrial and aquatic carbon processes have also been integrated into the Soil and Water Assessment Tool (SWAT) to simulate aquatic DOC dynamics (Du et al., 2019). These modules typically simulate individual reactions without considering multi-elemental thermodynamics and kinetics.

In this context, the recently-developed BFP model (Biogeochemical Reactive Transport - Flux - Penn State Integrated Hydrologic Modeling System, BioRT-Flux-PIHM) fills an important gap by incorporating coupled elemental cycling, stoichiometry, and rigorous thermodynamics and kinetics (Bao et al., 2017; Zhi et al., 2019). We used the BFP to address the question, *how do hydrology and T interact to determine rates of DOC production and export at the catchment scale?* We applied the BFP to a temperate forest catchment in the Susquehanna Shale Hills Critical Zone Observatory (SSHCZO) with extensive data. This small catchment (<0.1 km$^2$) has gentle topography with a network of shallow depressions or swales that have high SOC and deep soils (detailed in Section 2). It is underlain with only with one type of lithology (shale) and land use (forest), providing a useful testbed to evaluate biogeochemical and hydrological functions (Brantley et al., 2018). Previous lab and field work have identified non-chemostatic C-Q patterns of DOC at SSHCZO that are attributable to differences in the hydrologic connectivity of organic-rich soils during different flow conditions (Andrews et al., 2011; Herndon et al., 2015). SSHCZO has spatially-extensive and high-frequency measurements of soil properties, hydrology, and biogeochemistry (Brantley et al., 2018). These data facilitate detailed benchmarking of the BFP model and evaluation of processes controlling DOC production and export. We expected that *T* and soil moisture would drive DOC production in the soil, while DOC export and thus C-Q patterns would be most related to

hydrological connectivity. Therefore, we predicted that DOC production and export might be asynchronous (i.e. not happening at the same time) because they respond differently to changes in *T* and hydrology. Although soil respiration is an important process, this study focuses on the net production and export of DOC.

## 2. Methods

### 2.1. Study site: small catchment with an intermittent stream

The Shale Hills catchment is a 0.08 km$^2$, V-shaped, first-order watershed with an intermittent stream in central Pennsylvania. It is forested with coniferous trees and is situated on the Rose Hill Shale Formation. The annual mean air *T* is 9.8±1.9 $^\circ$C (±SD) and the annual mean precipitation is 1029±270 mm over the past decade. The watershed is characterized by large areas of swales and valley floors with deep and wet soils (Figure 1B). These lowland soils contain more SOC (~ 5% v/v) than the hillslopes and uplands (~ 1% v/v; Figure 1C).

We collected soil water DOC samples with lysimeters with a diameter of 5 cm installed at 10- or 20-cm intervals from the soil surface to a depth of hand-auger refusal, which varied from 30 to 160 cm depending on soil thickness. There were a total of six sampling locations (Figure 1B), including three at the south planar sites— valley floor (SPVF), midslope (SPMS), and ridgetop (SPRT)—and three at the swale sites—valley floor (SSVF), midslope (SSMS), and ridgetop (SSRT). No soil water DOC samples were collected at the north side of the catchment. Stream water DOC samples were collected daily in glass bottles at the stream outlet weir. All soil water and stream water DOC samples were filtered to 0.45 μm with Nylon syringe filters and were analyzed with a Shimadzu TOC-5000A analyzer (detailed in Andrews et al. (2011)). Real-time soil *T* (every 10 mins) was measured at the ridge top, midslope, and valley floor (squares in Figure 1B) using automatic monitoring stations at depths of ~ 0.10, 0.20, 0.40, 0.70, 0.90, 1.00 and 1.30 m (Lin and Zhou, 2008).

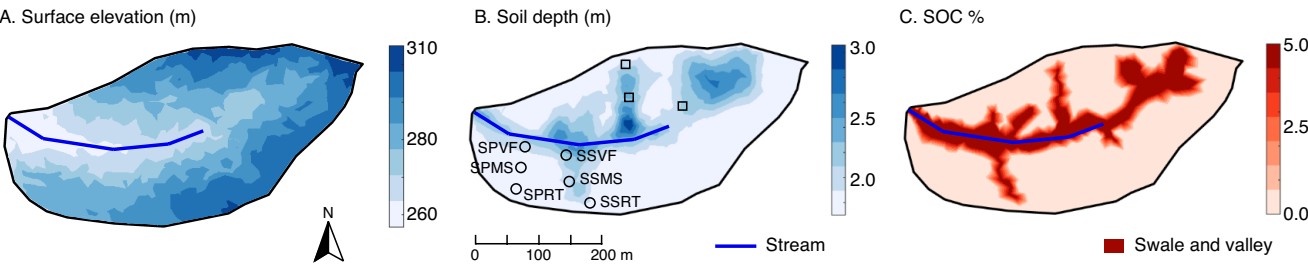

Figure 1. Attributes of the Susquehanna Shale Hills Critical Zone Observatory (SSHCZO): (A) surface elevation, (B) soil depth, and (C) soil organic carbon (SOC). Surface elevation was generated from LiDAR topographic data (criticalzone.org/shale-hills/data) while soil depths and SOC were interpolated using ordinary kriging based on field surveys with 77 and 56 sampling locations, respectively (Andrews

et al., 2011; Lin, 2006). The SOC distribution in Panel C is further simplified using the high, uniform SOC (5% v/v) in swales and valley soils based on field survey (Andrews et al., 2011). Swales and valley bottom were defined based on surface elevation through field survey and a 10-m resolution digital elevation model (Lin, 2006). Additional sampling instrumentation is shown in Panel B, including 6 soil water sites (circles) and 3 soil $T$ sites (squares).

## 2.2 The BFP model

BFP is the catchment reactive transport model of the general PIHM (Penn State Integrated Hydrologic Modeling System) family of code (Duffy et al., 2014). The code includes three modules (Figure 2): the surface hydrological module PIHM, the land-surface module Flux, and the multicomponent reactive transport module BioRT (Biogeochemical Reactive Transport). The code has been applied to simulate conservative solute transport, chemical weathering, surface complexation, and biogeochemical reactions at the catchment scale (Bao et al., 2017; Zhi et al., 2019; Li, 2019). Here we only introduce the salient features that are relevant to this study; readers are referred to earlier publications for further details. Flux-PIHM separates the subsurface flow into active interflow in shallow soil zones and groundwater flow deeper than the soil-weathered rock interface. The PIHM module simulates hydrological processes including precipitation, infiltration, surface runoff $Q_S$, soil water interflow (lateral flow) $Q_L$, and discharge $Q$ (Figure 2). The Flux module simulates processes including solar radiation and evapotranspiration. Flux-PIHM calculates water variables (e.g. water storage, soil moisture, and water table depth) in unsaturated and saturated zones and assumes a no-flow boundary at the soil-bedrock interface with high permeability contrast. In this version of Flux-PIHM, the deeper groundwater flow $Q_G$ is a separate input to the stream and is decoupled from the shallow soil water. This is supported by field data that shows negligible seasonal variation in groundwater chemistry (Jin et al., 2014; Thomas et al., 2013; Kim et al., 2018). The $Q_G$ is estimated using conductivity mass balance hydrograph separation (Lim et al., 2005).

The BioRT module takes in water calculated at each time step to simulate reactive transport processes. BFM discretizes the domain into prismatic elements and uses a finite volume approach considering the mass conservation governing equation for the reactive transport of a single solute $m$ is as follows:

$$V_i \frac{d(S_{w,i}\theta_i C_{m,i})}{dt} = \sum_{j=N_{i,1}}^{N_{i,x}} \left( A_{ij} D_{ij} \frac{C_{m,j} - C_{m,i}}{l_{ij}} - q_{ij} C_{m,j} \right) + r_{m,i}, \quad m = 1, np \quad (1)$$

where $i$ and $j$ represent the grid block $i$ and the neighboring grid $j$; the subscript $x$ distinguishes between flow in the unsaturated zone (infiltration and recharge) and saturated zone (recharge and lateral flow); $V$ is the total bulk volume (m³) of each grid block; $S_w$ is the soil moisture (m³ water/m³ pore volume); $\theta$ is porosity; $C$ is the aqueous

species concentration (mol/m$^3$ water); $t$ is time (s); $N$ is the index of elements sharing surfaces; $A$ is the grid interface area (m$^2$); $D$ is the diffusion/dispersion coefficient (m$^2$/s); $l$ is the distance (m) between the center of two neighboring grid blocks; $q$ is the flow rate (m$^3$/s); $r_m$ is the kinetically controlled reaction rates (mol/s) involving species $m$, which is the DOC production rate from SOC decomposition at the grid block $i$; and $np$ is the total number of independent solutes.

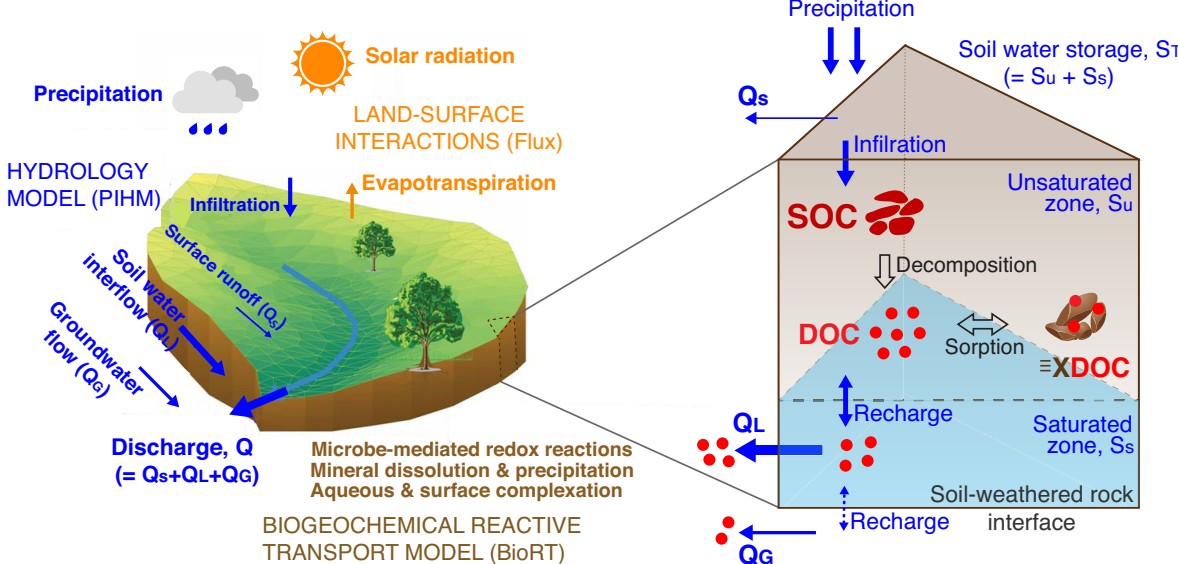

Figure 2. A schematic representation of major processes in the catchment reactive transport model BFP. Stream discharge $Q$ includes surface runoff $Q_S$, soil water interflow (lateral flow) $Q_L$, and groundwater flow $Q_G$. In the vertical direction, soil pores are not saturated with water in the shallow unsaturated zone and water flows vertically until it reaches the saturated zone where water forms interflows and moves laterally to the stream. Soil water total storage $S_T$ is the sum of water in the unsaturated ($S_u$) and saturated zones ($S_s$). Some water also recharges further into deeper groundwater. At SSHCZO, the interflow ($Q_L$) contributes about 90% to stream discharge, whereas the groundwater flow ($Q_G$) and surface runoff ($Q_S$) contributes about 8% and 2%, respectively on an annual basis. Within the soil zone, SOC decomposes and releases DOC, which also sorbs on the soil surface to become $\equiv XDOC$.

**DOC production and sorption.** In the model, DOC is produced by the decomposition of SOC via the kinetically-controlled reaction $SOC(s) \rightarrow DOC$. With abundant SOC and O$_2$ in soils serving as electron donors and acceptors, a typical dual Monod kinetics can be simplified into zero-order kinetics with additional temperature and soil moisture dependence:

$$r_p = kAf(T)f(S_w) \qquad (2)$$

where $r_p$ is the local DOC production rate in individual grids ($r_m$ in Eq. 1, $m$ is DOC); $k$ is the kinetic rate constant of net DOC production (= $10^{-10}$ mol/m$^2$/s) (Zhi et al., 2019; Wieder et al., 2014); and $A$ is a lumped "surface area" term (m$^2$, = $2.5 \times 10^{-3}$ m$^2$/g $\times$ g of SOC mass) that quantifies SOC content and biomass abundance (Chiou et al., 1990; Kaiser and Guggenberger, 2003; Zhi et al., 2019). The functions $f(T)$ and $f(S_w)$ describe the rate dependence on soil $T$ and moisture, respectively. $f(T)$ follows a widely-used Q$_{10}$-based formation: $f(T) = Q_{10}^{|T-10|/10}$, where $Q_{10}$ quantifies the rate increases with $T$, with the number 10 in the superscript referring to $T$ of 10 °C (Davidson and Janssens, 2006). $Q_{10}$ in the base case is set at 2.0, within the typical range of 1.2-3.8 for forest ecosystems (Liu et al., 2017). The $f(S_w)$ has the form $f(S_w) = (S_w)^n$ in the base case, where $n$ is the saturation exponent with a value of 1.0, which is within the typical range of 0.75-3.0 for most soils (Yan et al., 2018; Hamamoto et al., 2010). The dependence of production rates on soil $T$ and moisture have been described with multiple forms in existing studies (Davidson and Janssens, 2006; Yan et al., 2018) and will be further explored through sensitivity analysis, as detailed in Section 2.6. SOC content typically decreases with depth (Billings, 2018; Bishop et al., 2004), though the specific pattern may vary with soil texture, landscape position, vegetation, and climate (Jobbagy and Jackson, 2000). The depth function of SOC at Shale Hills has been observed to be exponential (Andrews et al., 2011), which is typical of many soils (Billings et al., 2018; Currie et al., 1996). To take this into account, we use the equation $C_d(z) = C_0 exp\left(-\frac{z}{b_m}\right)$, where $C_d$ is SOC at depth $z$ below the surface; $C_0$ is the SOC level at the ground surface and $b_m$ quantified the decline rate with depth, set here to a value of 0.3 (Weiler and McDonnell, 2006).

DOC produced from SOC can also sorb on soils via the reaction $\equiv X + DOC \leftrightarrow \equiv XDOC$, where $\equiv X$ and $\equiv XDOC$ represents the functional group that can sorb DOC and the functional group with sorbed DOC, respectively (Rasmussen et al., 2018). This reaction is considered fast and is thermodynamically-controlled with an equilibrium constant $K_{eq}$ that links the activity (here approximated by concentrations) of the three chemicals via $K_{eq} = \frac{[\equiv XDOC]}{[\equiv X][DOC]}$. The DOC concentrations calculated from Eq. (1) are used to calculate the concentrations of $\equiv X$ and $\equiv XDOC$. The $K_{eq}$ value represents the thermodynamic limit of the sorption, i.e., the sorption affinity of the soil for DOC. It depends on temperature but also soil properties such as the content of clay and iron oxides (Kaiser et al., 2001; Conant et al., 2011). A $K_{eq}$ value of $10^{0.2}$ was obtained by fitting the stream and soil water DOC data (detailed in Section 2.4). The sum of $[\equiv X]$ and $[\equiv XDOC]$ represents the sorption capacity of the soil with a value ranging from $4.0 \times 10^{-5}$ - $6.0 \times 10^{-5}$ mol/g soil at Shale Hills (Jin et al., 2010; Li et al., 2017), depending on the mineralogy.

**2.3 Domain setup**

BFP is a 2.5D model with full discretization in the horizontal directions and partial discretization in the vertical direction with three layers: ground surface, unsaturated, and saturated zones. The study watershed is discretized into 535 prismatic land elements and 20 stream segments using PIHMgis (http://www.pihm.psu.edu/pihmgis_home.html), a GIS interface that is tightly coupled to BFP. The land elements are unstructured triangles with mesh sizes varying from 10 to 100 m. The simulation domain is set up using national datasets, including the USGS National Elevation Dataset for topography, the National Land Cover Database for land cover, the National Hydrography Dataset, the North American Land Data Assimilation Systems phase 2 (NLDAS-2) for hourly meteorological forcing, and the Moderate Resolution Imaging Spectroradiometer (MODIS) for leaf area index every eight days. In addition, extensive characterization and measurement data at Shale Hills have been used to define soil depth and soil mineralogical properties such as surface area, and ion exchange capacity that are heterogeneously distributed across the catchment (Andrews et al., 2011; Lin, 2006; Jin and Brantley, 2011; Jin et al., 2010; Shi et al., 2013) (criticalzone.org/shale-hills/data/). Other soil matrix properties include conductivity, porosity, and van Genuchten parameters. Soil macropores such as cracks, fractures, and roots can generate preferential flows. Their properties are represented using the area macropore fraction, depth, and conductivities. They are parameterized based on values quantified in previous studies at Shale Hills (Shi et al., 2013; Lin, 2006), shown in Figure S1 and Table S1.

Based on field measurements, SOC content in swales and valley is relatively high (Andrews et al., 2011) and was set at 5 % (v/v solid phase) compared to 1% in the rest of the catchment (Figure 1C). The clay minerals were set a value of 23% (v/v solid phase) along the ridgetop and increased to 33% at valley floor (Jin et al., 2010; Li et al., 2017). The input DOC concentration in rainfall and groundwater (below soils) were set at reported medians of 0.6 and 1.2 mg/L, respectively (Andrews et al., 2011; Iavorivska et al., 2016), as high frequent DOC observations for rainfall and groundwater are not available at the site. The initial DOC concentration in soil water was set at 2.0 mg/L, the average concentration from the six field sampling locations.

**2.4 Model calibration**

We used stream (daily) and soil pore water (biweekly) DOC concentration data during April-October 2009 for model calibration and the year 2008 as spin-up until a "steady state" for both water and DOC. The

"steady state" here refers to the state of mass balance that the difference of overall annual input, output and
changes of mass stored within the catchment is less than 5% of the corresponding overall annual input. The water
input is precipitation while its output is the sum of ET and discharge. The DOC mass input is from rainfall, SOC
decomposition, and groundwater while the DOC mass export at the stream outlet is the output. The model
performance was evaluated using the monthly Nash-Sutcliffe efficiency (NSE) (Nash and Sutcliffe, 1970) that
quantified the relative magnitude of residual variance of modeling output compared to measurements. The general
satisfactory range for monthly-average outputs for hydrological models is NSE > 0.5 (Moriasi et al., 2007) and
we used similar standards for biogeochemical solutes (Li et al., 2017). To reproduce the DOC data, we first set
the SOC surface area $A$ using a literature range of $10^{-3}$-$10^{0}$ $m^2$/g (Zhi et al., 2019; Chiou et al., 1990; Kaiser and
Guggenberger, 2003). We also set $K_{eq}$ using a literature range of $10^{0}$-$10^{1}$ (Oren and Chefetz, 2012; Ling et al.,
2006). Once the simulated output captured the temporal trend of data, we refined $Q_G$ based on the estimation from
hydrograph separation (Figure S2) to capture the peaks of stream DOC concentration, especially under low
discharge periods. Because not all soils are in contact with water, the calibrated surface area represents the
effective solid-water contact area in a heterogeneous subsurface, and is orders of magnitude lower than the
reported SOC surface areas from laboratory experiments (Kaiser and Guggenberger, 2003). The calibrated
hydrological parameters are mostly from Shi et al. (2013), except groundwater estimation. With the overall
groundwater flow estimated in Li et al. (2017), groundwater estimates were further refined by calculating average
groundwater fluxes in wet and dry periods using conductivity mass-balance hydrograph separation (Lim et al.,
2005) and then by reproducing the stream DOC concentration. In other words, stream and groundwater chemistry
data helped constrain the groundwater flow into the stream.

**2.5 Quantification of water and DOC dynamics at the catchment scale**

**Hydrological connectivity.** Modeled spatial and temporal outputs of saturated soil water storage were used to
quantify hydrological connectivity *Width* defined as the average width of catchment in the direction perpendicular
to the stream (230 m), The term $I_{cs}$/*Width* quantifies the average proportional width of the catchment connected
to the stream (e.g., $I_{cs}$/*Width* = 0.10, 0.35, and 0.70 in Figure S3). Depending on the catchment geometry and
extent of connectivity, $I_{cs}$/*Width* may vary from 0 to 1.0. Note that $I_{cs}$/*Width* may exceed 1.0 for catchments with
the length >> width under extreme precipitation events. High $I_{cs}$/*Width* value (i.e., high hydrological connectivity)
indicates that a large catchment area is connected to the stream. To determine whether two land elements are
hydrologically connected, the spatial distribution of saturated water storage was used following $I_{cs}$ =

$\int_0^\infty \tau(h)dh$ based on an algorithm in the literature (Allard, 1994; Western et al., 2001; Xiao et al., 2019). Here

$\tau(h)$ is the probability of two grid blocks being connected at a separation distance of $h$, in which a threshold is defined as the 75th percentile of saturated storage over the whole catchment. Note that $I_{cs}/Width$ here only quantifies the hydrological connectivity in soils and does not reflect the groundwater in shallow aquifers below the soil-bedrock interface.

**DOC at the catchment scale.** At the catchment scale we differentiate two different rates (mg/d): DOC production and export. The production rate $R_p$ is the sum of the local DOC production rate $r_p$ in individual grid blocks (Eq. (2)) across the whole catchment, including those produced and the sorbed on soils. The export rate $R_e$ is calculated as the product of discharge and DOC concentration at the stream outlet. Total stored DOC is the difference between stream output and input from production, rainfall, and groundwater. The DOC input from the rainfall $R_r$

(mg/d) is the precipitation rate (m/d) times the rainfall DOC concentration ($6.0\times10^{-4}$ mg/m$^3$ = 0.6 mg/L$\times10^{-3}$ L/m$^3$) and the catchment drainage area (m$^2$). The DOC input from groundwater $R_g$ (mg/d) is the total groundwater influx (groundwater flow rate × catchment drainage area) times the groundwater DOC concentration (1.2 mg/L).

C-Q patterns were quantified using two complementary approaches: the power law equation $C = aQ^b$ (Godsey et al., 2009) and the ratio of coefficient of variations of DOC concentration and discharge $\frac{CV_{[DOC]}}{CV_Q}$

(Musolff et al., 2015). We used both methods because the slope of the power law equation does not account for the goodness-of-fit of the C-Q pattern itself. For example, a slope of $b = 0$ would be considered chemostatic (i.e. relatively small variation of concentration compared to discharge), although high variability in the solute concentration would actually render the behavior chemodynamic (i.e., solute concentrations are sensitive to changes in discharge) (Musolff et al., 2015). We considered two general categories based on these metrics

(Godsey et al., 2009; Underwood et al., 2017; Musolff et al., 2015): If $b$ fall between -0.2 and 0.2 and $\frac{CV_{[DOC]}}{CV_Q} <<$

1, C-Q patterns were considered chemostatic; Values of $|b| > 0.2$ or $\frac{CV_{[DOC]}}{CV_Q} \geq 1$, indicated a chemodynamic behavior. In the chemodynamic category, values of $b>0.2$ indicate flushing, while values of $b < -0.2$ indicate dilution. We used the Matlab curve-fitting toolbox to obtain both the best fit model parameters and the goodness-of-fit measures, including confidence intervals and the $R^2$ statistic.

**2.6 Sensitivity analysis**

We used a sensitivity analysis to explore the influence of soil $T$ and moisture. The $Q_{10}$ in $f(T) = Q_{10}^{|T-10|/10}$ was explored using a minimum value of 1.0 (i.e. no dependence on $T$) and a maximum value of 4.0 (Davidson and Janssens, 2006) (Figure S4A), i.e. $f(T) = 1$ and $f(T) = 4^{|T-10|/10}$. The rate dependence on soil moisture was explored using the base case $f_1(S_w) = (S_w)^n$ (increase behavior), and three additional functions ($f_2$, $f_3$, and $f_4$) representing the most commonly observed forms (Figure S4B), including decrease behavior, constant behavior, and threshold behavior (Gomez et al., 2012; Yan et al., 2018):

$$\text{Decrease-behavior function} \quad f_2(S_w) = (\frac{1-S_w}{0.6})^{0.77} \tag{3}$$

$$\text{Constant-behavior function} \quad f_3(S_w) = 0.65 \tag{4}$$

$$\text{Threshold-behavior function} \quad f_4(S_w) = \begin{cases} (\frac{S_w}{0.7})^{1.5} & S_w \le 0.7 \\ (\frac{1-S_w}{1-0.7})^{1.5} & S_w > 0.7 \end{cases} \tag{5}$$

The constants in Eq. (3)-(5) were selected to ensure similar averages of $f(S_w)$ across the whole $S_w$ range such that trajectories rather than absolute values of $f(S_w)$ were compared (Figure S4B).

We also tested the sensitivity of DOC sorption onto soils by changing $K_{eq}$ between values of 0 (no sorption), $10^{0.5}$ and $10^{1.0}$. As an important source of stream water and DOC, the sensitivity of C-Q patterns and $R_e$ to changes in groundwater was also tested with groundwater flow contribution and DOC concentration. The groundwater flow rates were varied from negligible ($Q_G = 0$) to 2.5 times of those at the base case ($Q_G = 3.3\times10^{-4}$ and $1.0\times10^{-4}$ m/day the wet and dry periods, respectively). The corresponding fractions ($Q_G/Q$) of groundwater flow to the total annual discharge for the two cases were 0 and 18.8%, respectively. The groundwater DOC concentration ($DOC_{GW}$) was varied by two orders of magnitude (0.12 mg/L and 12.0 mg/L). We compared results from these analyses to the base case for which the groundwater contributed to a 7.5% of the total annual streamflow at 1.2 mg/L.

## 3. Results

### 3.1. Dynamics in the base case

**Water dynamics.** The total precipitation from 1 April 2009 to 31 March 2010 was 1,130 mm. Measured stream discharge at Shale Hills (Figure 3) was highly responsive to intense precipitation events and was high ($\sim 10^{-2}$ m/day) in spring and fall compared to summer with high soil $T$ and high ET ($\sim 10^{-5}$ m/day). The model captured the temporal dynamics of daily discharge, ET, and soil $T$ with an NSE value of 0.68, 0.72, and 0.62, respectively

(Figure 3A-B). The model estimated that 47.5% of annual precipitation contributed to discharge while the rest to ET. The stream discharge has three components, including surface runoff $Q_S$, soil water interflow $Q_L$ (lateral flow) and groundwater flow $Q_G$ from shallow aquifers that interact with the stream (Figure 2). On average, lateral flow $Q_L$ is about 90.2% and surface runoff $Q_S$ is about 2.3%. Following the conductivity mass-balance hydrograph separation (Lim et al., 2005), $Q_G$ was estimated as $1.3 \times 10^{-4}$ and $4.0 \times 10^{-5}$ m/day for the wet and dry periods (August

– September), equivalent to 6.9% and 42.2% of average stream discharge in the corresponding times, respectively. Overall $Q_G$ accounted for ~ 7.5% of the annual $Q$, similar to previously reported values (Li et al., 2017; Hoagland et al., 2017). In the dry months from August to September, the stream was almost dry with no visible flow and the relative contribution of groundwater to discharge was comparable to that of $Q_L$ (Figure 3B).

      The unsaturated water storage $S_u$ was typically more than 10 times larger than the saturated storage $S_s$

such that the $S_T$ and $S_u$ curves almost overlapped (Figure 3C). $S_s$ was negligible in the dry period (close to 0 m), contributing negligibly to the stream. Hydrological connectivity ($I_{cs}/Width$) covaried with $S_s$ but showed significant temporal fluctuations. High summer ET drove the catchment to drier conditions, therefore decreasing the connectivity to the stream.

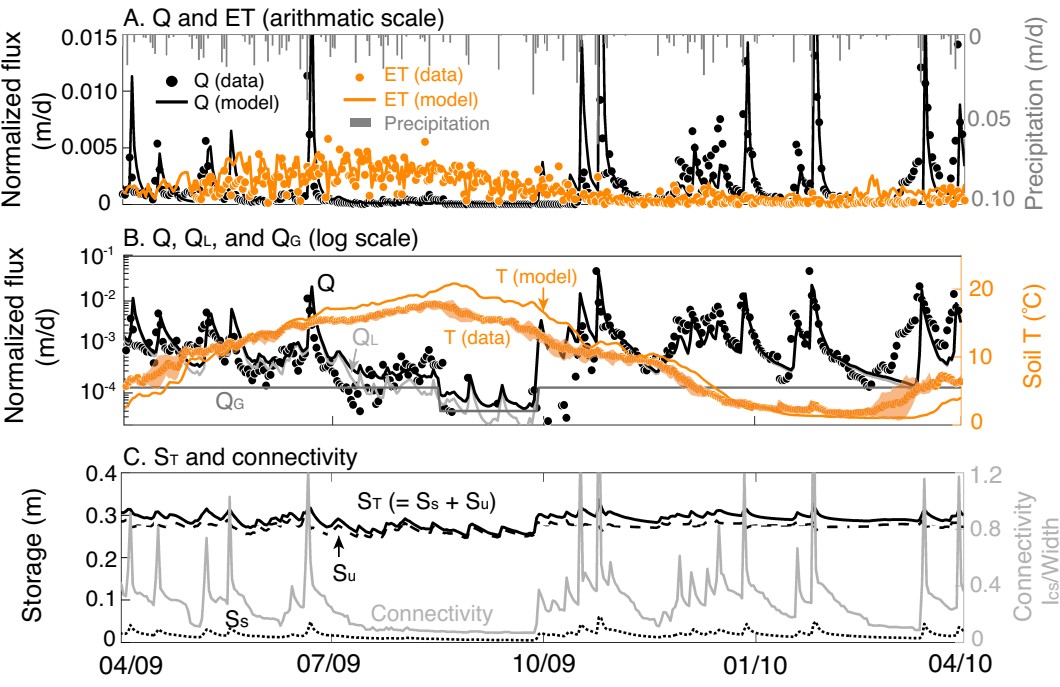

Figure 3. Temporal dynamics of (A) daily precipitation, stream discharge $Q$, and evapotranspiration ET on an arithmetic scale; (B) stream discharge $Q$, soil water interflow $Q_L$, and groundwater $Q_G$ on a

logarithmic scale with soil $T$ on an arithmetic axis on the right; (C) soil water storage $S_T$ (= unsaturated water storage $S_u$ + saturated water storage $S_s$) and hydrological connectivity $I_{cs}/Width$. The yellow dots in Panel B represent the average soil $T$ from 3 sampling locations (square symbols in Figure 1B) while the shaded zone is the measurement variation. $Q$ was highly responsive to intense precipitation events in spring and winter. Note high soil $T$, high $ET$, low $S_s$, and low $I_{cs}/Width$ during July-August 2009. Stream discharge was primarily comprised of $Q_L$, except in July-October when the relative contribution of $Q_G$ increased.

**Temporal patterns of DOC concentrations**. The model captured the general trend of stream DOC (NSE = 0.55 for monthly DOC concentration; Figure 4). A temporal pattern emerged from changes in the relative contribution of soil water $Q_L$ and groundwater $Q_G$ to stream discharge $Q$ through time. Under dry conditions (e.g., $Q < 1.0 \times 10^{-4}$ m/day), $Q_G$ contributed substantially to $Q$ (~32-71%; Figure 3), and stream DOC concentration reflected the mixing of groundwater and soil water (Figure 4A), with a contribution from groundwater DOC of 7-17%. Under wet conditions, stream DOC concentration overlapped with soil water DOC concentration (light blue line in Figure 4). Only ~1-8% of stream DOC was sourced from groundwater at these times.

The temporal dynamics of field-measured soil water DOC at local scales showed relatively less temporal variation than stream DOC (Figure 4B-G), and local soil pools were not always hydrologically connected to the stream. The simulated soil water DOC at local scales captured this less-variation trend and the overall model performance was acceptable (i.e., NSE >0.5), though the goodness-of-fit was lower for some locations (e.g. NSE value of 0.36 (SPRT), 0.42 (SPMS), 0.60 (SPVF), 0.46 (SSRT), 0.40 (SSMS), and 0.51 (SSVF)). This discrepancy between overall and partial model performance may be due to local variation in soil properties and organic carbon content for which we do not have detailed information. Although the model explicitly considered spatial heterogeneities such as topography and soil properties, averaged values represented grid sizes from 10 to 100 m, and this local scale was large compared to field sampling size (e.g., lysimeters with a diameter of 5 cm). Geochemical processes are sensitive to local properties, including SOC%, SOC surface area and sorption sites, while the representation of these properties was based on a few measurements that were only coarsely defined as ridgetop, midslope, and valley floor.

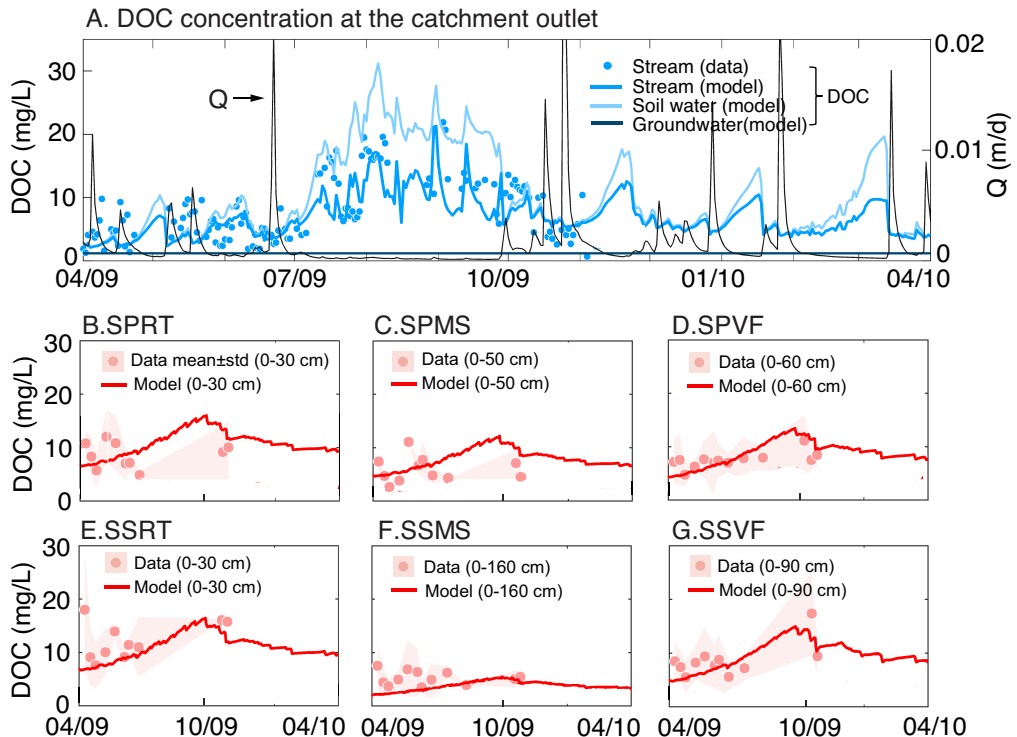

Figure 4. (A) Temporal dynamics of measured and simulated stream DOC concentration as well as groundwater and soil water DOC. Stream DOC (bright blue line) was from the soil water (light blue line) and groundwater $Q_G$ (dark blue line). Under low discharge conditions (e.g., July-September), $Q_G$ contributed a larger proportion of discharge and stream DOC was more similar to groundwater DOC. Under wet conditions, stream DOC resembled soil water DOC from $Q_L$. (B)-(G) Local soil water DOC concentration for the 6 sampling locations shown in Figure 1B, including 3 planar (panels B-D) and 3 swale locations (panels E-G). The mean ± standard deviation for each location was calculated based on measurements at different depths with 10- or 20-cm intervals from the soil surface down to depth of hand-auger refusal.

**Spatial patterns and mass balance.** There were differences in the spatial patterns of soil environmental variables, DOC production, and soil water DOC concentration between May (wet), August (dry) and October (wet after dry) (Figure 5). In May, the average soil $T$ was around 12 °C with relatively minor variations (< 3 °C) across the catchment. Most flow-convergent areas (valley and swales) were well connected to the stream and had high water content (Figure 5B-C). The distribution followed that of SOC (Figure 1C) and water content (Figure 5B), with high $r_p$ and soil water DOC concentration in swales and valley. Low $r_p$ in relatively dry planar hillslopes and uplands led to low soil water DOC concentration. In August, the average soil $T$ increased to around 20°C. The hydrologically-connected zones shrank to the immediate vicinity of the stream, but $r_p$ increased by 2-fold from

May. Simulated soil water DOC concentration increased by a factor of 2 across the whole catchment, especially in hillslope and uplands at the north side of the catchment, partly because the produced DOC was trapped in low soil moisture areas that were not hydrologically connected to the stream. This indicates that DOC samples collected at the south side may not well represent the DOC dynamics of the entire catchment, especially in the summer and fall dry months. In October, $r_p$ decreased as soil cooled down, but increased precipitation and decreased ET expanded the hydrologically connected zones beyond swales and valley to include more of the upland hillslopes (Figure 5C). The increase in hydrological connectivity favored the desorption and the flushing of stored DOC, although the soil water DOC concentration remained high because of the large store of sorbed DOC produced during the antecedent dry times.

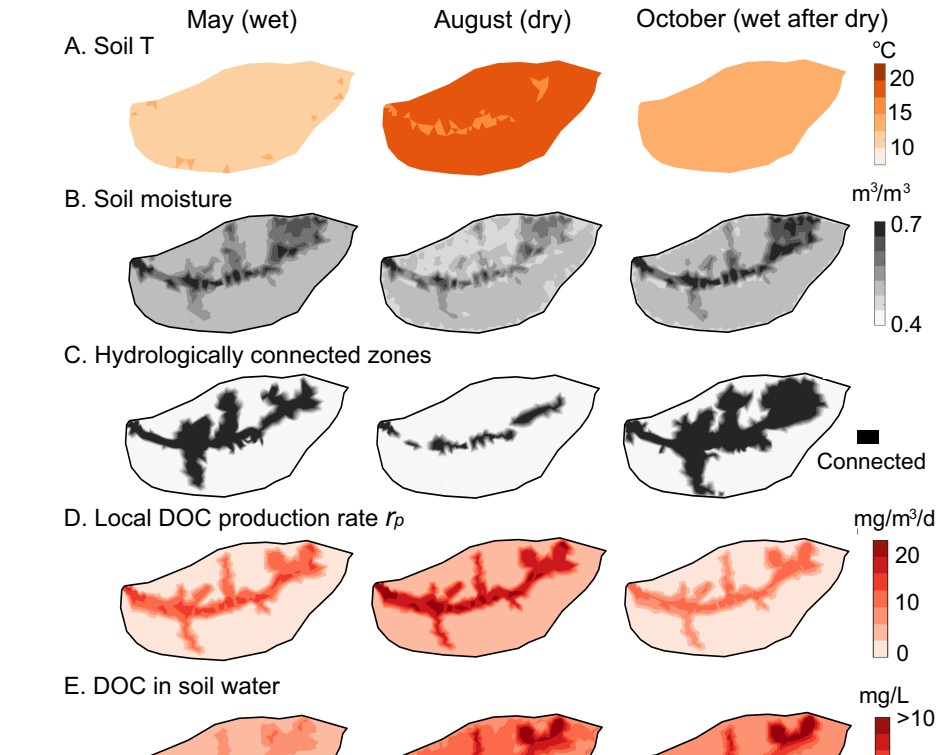

Figure 5. Spatial profiles in May (wet), August (dry), and October (wet after dry) of 2009: (A) soil $T$, (B) soil moisture, (C) hydrologically connected zones, (D) local DOC production rates $r_p$ and (E) soil water DOC concentration. The soil DOC and $r_p$ are high in swales and the main valley that have relatively high soil water and SOC content (Figure 1C). Although water content in August is relatively low

compared to May and October, higher soil $T$ leads to higher $r_p$, with most DOC production and accumulation in zones that are disconnected to the stream.

Figure 6 shows the catchment-scale DOC production and export rates and mass balance. Generally, the daily $R_p$ (5.1 ×10$^5$ mg/d) was greater than the daily $R_r$ from rainfall (1.6×10$^5$ mg/d) or groundwater $R_g$ (1.2×10$^4$ mg/d). During storm events, $R_r$ occasionally exceeded $R_p$. $R_p$ was generally high in summer, despite low water storage. Export rate $R_e$ did not follow the temporal patterns of either the total input rate ($R_p+R_r+R_g$) or $R_p$. Instead, it primarily followed the discharge patterns: large rainfall events exported disproportionally high DOC, leading to abrupt drops in DOC mass in the catchment. From the wet to dry period, as water levels dropped, DOC accumulated within the catchment (Figure 5E, May to August). During the dry-to-wet transition, as the catchment became wetter, the contributing areas expanded to uplands and the DOC was flushed out, reducing the overall DOC soil pool to much lower values (Figure 5E, August-October). The DOC mass storage increased 1.8×10$^6$ mg over the year, about 1.0% of the overall DOC production, which indicated a general mass balance at the catchment scale.

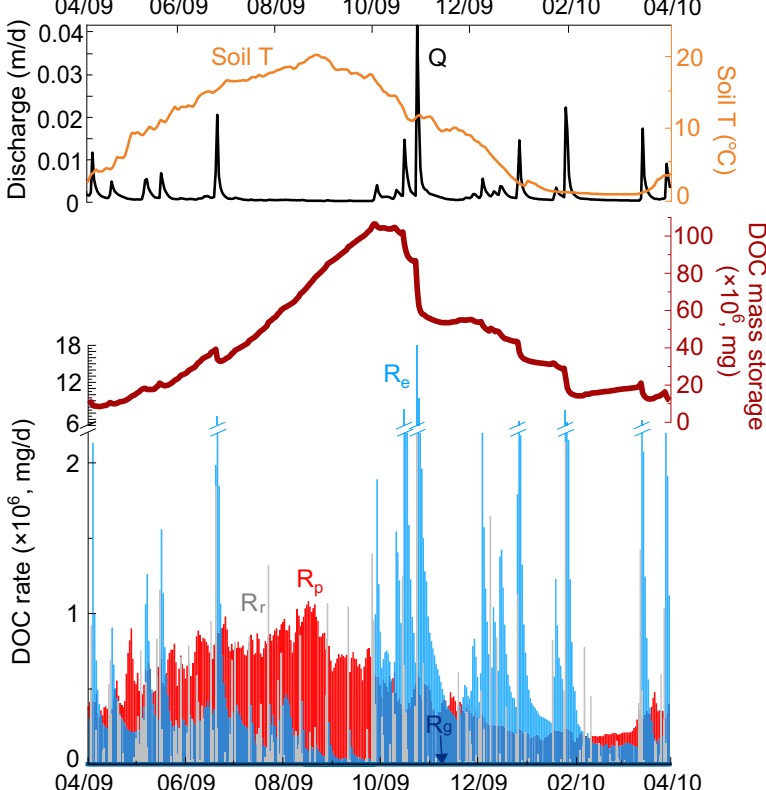

Figure 6. Temporal dynamics of DOC storage, influent rate (rainfall $R_r$, groundwater $R_g$, production $R_p$) and outflow rate (effluent $R_e$) at the catchment scale. The stored DOC mass (dark red line) was calculated as (DOC influent rate - outflow rate) × time. The temporal $R_e$ dynamics mostly followed the trend of discharge (black line, top panel) while $R_p$ mostly followed the trend of soil $T$ (orange line, top panel).

**C-Q patterns and rate dependence.** The C-Q relationships showed a slightly positive correlation at low $Q$ followed by a negative correlation at higher $Q$ (Figure 7A). The simulated C-Q relationship captured this trend but overestimated the positive relationship at low $Q$. The simulated C-Q relationships showed a general dilution behavior with the C-Q slope $b = -0.23$ and $\frac{CV_{[DOC]}}{CV_Q} = 0.22$, consistent with the general pattern exhibited in the field data (Figure 7A). This C-Q pattern can be explained by the dynamics of different water sources with different DOC contributing to the stream. At low discharges ($< 1.8\times10^{-4}$ m/d) with small water storage (0.25-0.28 m) and connectivity ($I_{cs}/Width$ <0.1) (Figure 7B), the stream DOC was a mix of organic-poor groundwater and organic-rich swales and valley floor zones. As connectivity and discharge increased and the stream expanded, the contribution of organic-rich swales increased, elevating DOC concentration to its maximum. At even wetter conditions with connectivity exceeding 0.1, the contribution from planar hillslopes and uplands with lower DOC concentration increased, diluting the organic-rich DOC from swales and valley. Daily $R_e$ correlated positively with $S_T$, hydrological connectivity and $Q$, and increased by two orders of magnitude as $Q$ rose by three order of magnitude. The variation of daily $R_p$ with $Q$ was small ($10^5$ -$10^6$ mg/d) compared to that of $R_e$ (Figure 7B).

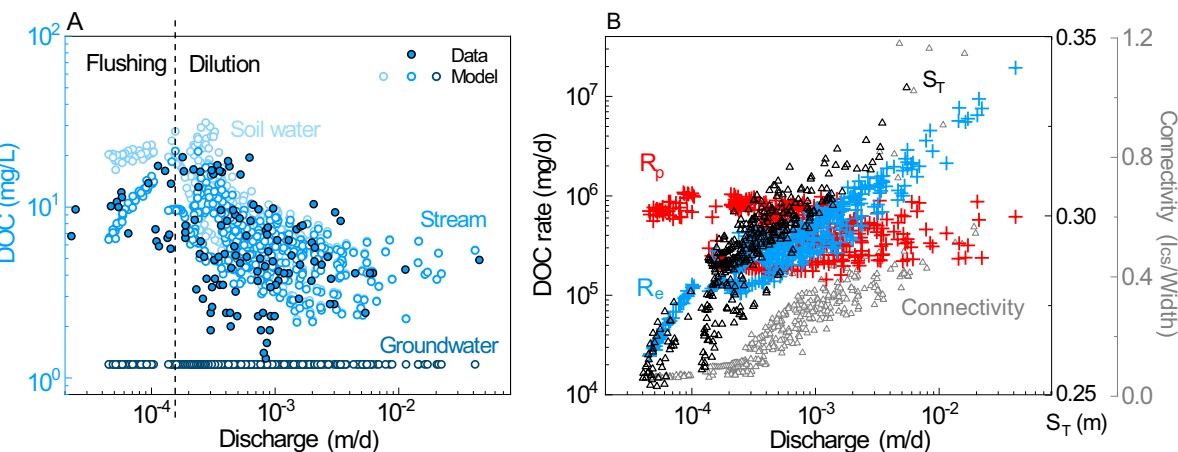

Figure 7. Relationships of daily discharge ($Q$) with: (A) stream DOC concentration; open circles are simulations; filled circles with a black outline are data; (B) soil water storage $S_T$, connectivity ($I_{cs}/Width$), and catchment-scale DOC export rate $R_e$, and DOC production rate $R_p$. At low $Q$, the stream water

transitioned from organic-poor groundwater to organic-rich water from valley floor and swales, leading to a flushing (positive) pattern. At higher $Q$, the stream water shifted from organic-rich soil water from swales and valley to lower DOC water from planar hillslopes and uplands, decreasing stream DOC concentration and resulting in a dilution C-Q pattern. $R_e$ increased by two orders of magnitude with increasing $Q$, while $R_p$ varied within an order of magnitude.

Values of $R_p$ depended more on soil $T$ than soil water storage and hydrological connectivity ($I_{cs}/Width$). In contrast, $R_e$ increased with soil water storage $S_T$ but notably decreased with soil $T$ (> 17 ºC) due to the low discharge during the hot and dry summer.

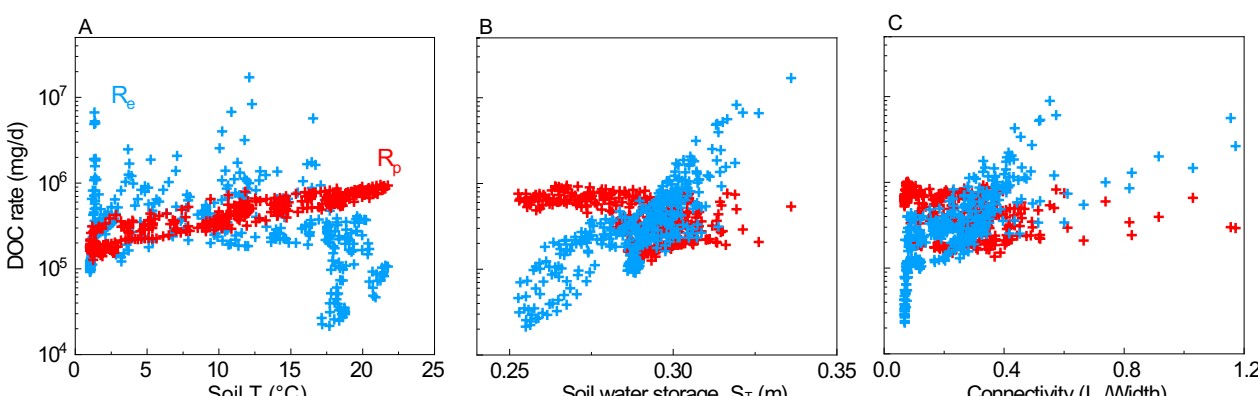

Figure 8. Catchment-scale DOC production rate $R_P$ and export rate $R_e$ as a function of (A) soil $T$, (B) soil water storage $S_T$, and (C) hydrological connectivity ($I_{cs}/Width$). Cross symbols are daily values in the base case. $R_p$ increased with soil $T$ and decreased slightly with $S_T$ and connectivity. In contrast, $R_e$ increased with $S_T$ and connectivity but decreased with soil $T$. $R_e$ tended to decrease with soil $T$ in the hot, dry summer because of low discharge in that period.

## 3.2. Sensitivity analysis

**Control of temperature, soil moisture, and sorption on DOC production and export.** Higher $Q_{10}$ values in $f(T)$ leads to more pronounced seasonality in $R_p$ (Figure 9A). The $R_p$ for $Q_{10}$=4.0 was more than 4 times higher than that of $Q_{10}$=1.0 in summer, and much lower in winter with low soil $T$ (< 10 °C). In contrast, the temporal pattern of $R_e$ almost overlapped at different $Q_{10}$ values, and mostly followed the discharge dynamics (black line in Figure 9). Daily $R_p$ varied only slightly (within 15%) with different $f(S_w)$ (Figure S4B), while $R_e$ almost did not change (Figure 9B). Though we varied $Q_{10}$ from 1.0 to 4.0 in $f(T)$, it is worth noting that varying kinetic rate constant, SOC surface area, volume fraction, and biomass amount could have similar effects (not shown here) because they are all multiplied in Eq. (2).

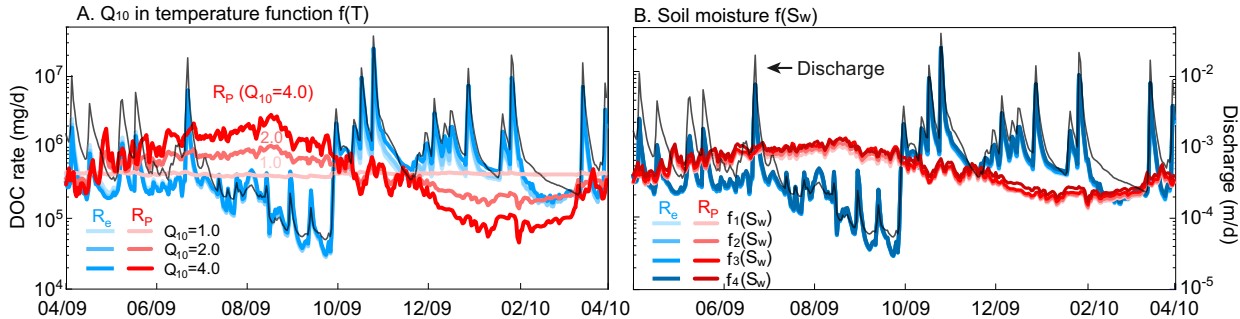

Figure 9. Sensitivity analysis of temporal DOC rates for (A) soil temperature f(T) and (B) soil moisture f($S_w$). Varying $Q_{10}$ value in f(T) had a larger influence on $R_p$ than varying f($S_w$). Neither f(T) nor f($S_w$) had a significant influence on $R_e$. Instead, $R_e$ mostly followed the temporal trend of discharge, indicating the predominant control of hydrological conditions.

Simulations showed that strong DOC sorption ($K_{eq} = 10^{1.0}$) did not change $R_p$ but lowered stream DOC concentration and resulted in smaller $R_e$ (Figure 10A). DOC sorption had little impact on $R_p$ but strong sorption decreased the magnitude of $R_e$ by 10-69%. The sorbed DOC concentration differed by more than a factor of 3, with more sorbed DOC with larger $K_{eq}$ values (Figure 10B). Large amounts of sorbed DOC persisted until early fall, when large rainfall events flushed out sorbed DOC and reduced DOC storage (Figure 6). This means that catchments can store large quantities of DOC, the specific amount of which depends on sorption capacity.

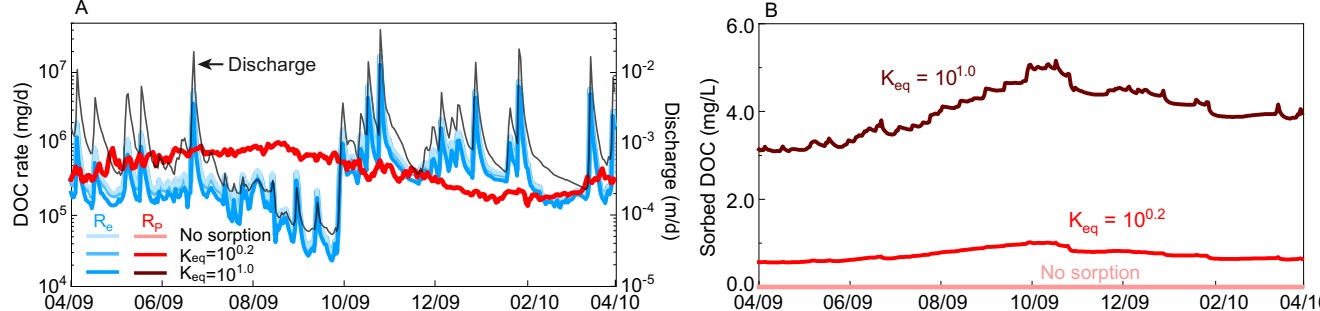

Figure 10. Sensitivity analysis of sorption equilibrium constant $K_{eq}$ on (A) $R_p$ and $R_e$ and (B) DOC sorbed on soils averaged at the catchment scale. High $K_{eq}$ led to more DOC sorbed on soils and therefore lower $R_e$. However, $R_e$ showed similar temporal patterns regardless of $K_{eq}$.

Varying DOC production kinetics did not change the overall C-Q patterns although the magnitude of overall dilution changed slightly in cases with different $f(T)$ and $K_{eq}$ (Figure 11). High $Q_{10}$ values in $f(T)$ led to less dilution, due to more accumulated soil DOC in the dry period (low discharge) and thus more DOC flushing as discharge increased in the dry-to-wet period. High $K_{eq}$ resulted in less dilution as higher sorption capacity acts as a stronger buffer to compensate the concentration variations.

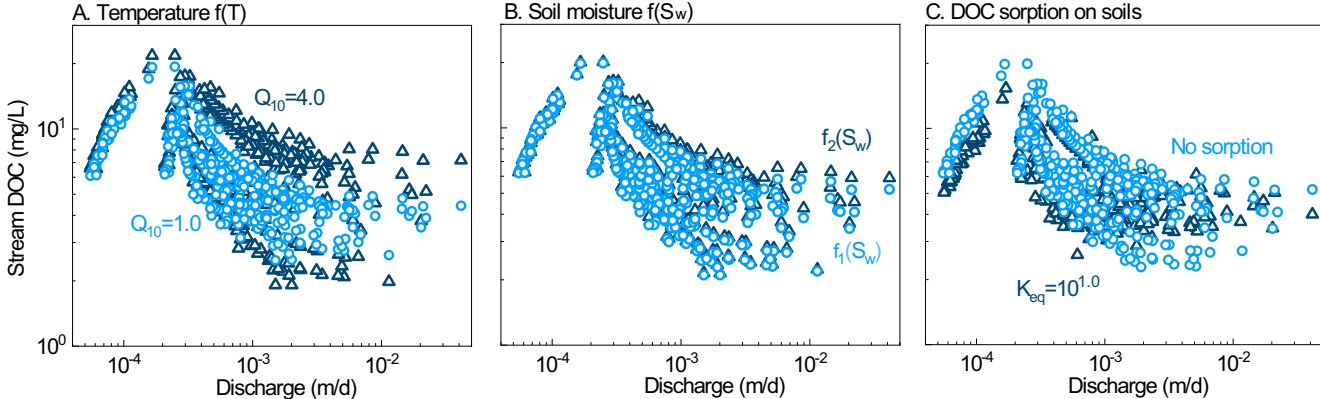

Figure 11. C-Q relationships under different (A) f(T), (B) f(Sw), and (C) sorption equilibrium constants $K_{eq}$ for the two extreme cases. The C-Q patterns were similar in all cases, although the extent of dilution slightly changed. This indicates potentially factors other than reaction kinetics and thermodynamics that regulate C-Q patterns.

**Groundwater control on DOC export.** As shown in Figure 12, changing groundwater volume contribution to stream (GW) had more significant impacts on the dynamics of $R_e$ than changing groundwater DOC concentration (DOC$_{GW}$), especially at low discharges ($Q < 1.8\times10^{-4}$ m/d). Increasing GW contribution from no GW to 2.5GW (i.e. 18.8%) lowered stream DOC at low discharges, shifting the C-Q pattern from overall dilution (or chevron pattern) to overall flushing (or flushing until stable). More specifically, the threshold that separated distinct phases of these segmented C-Q responses (Figure 12A2) shifted from $Q = 1.8\times10^{-4}$ m/d to about $1.0\times10^{-3}$ m/d. This reflects the relative groundwater contribution to streamflow for each case. In contrast, varying groundwater DOC concentration (DOC$_{GW}$) by 2 orders of magnitude while keeping the same groundwater contribution (GW) did not change C-Q pattern.

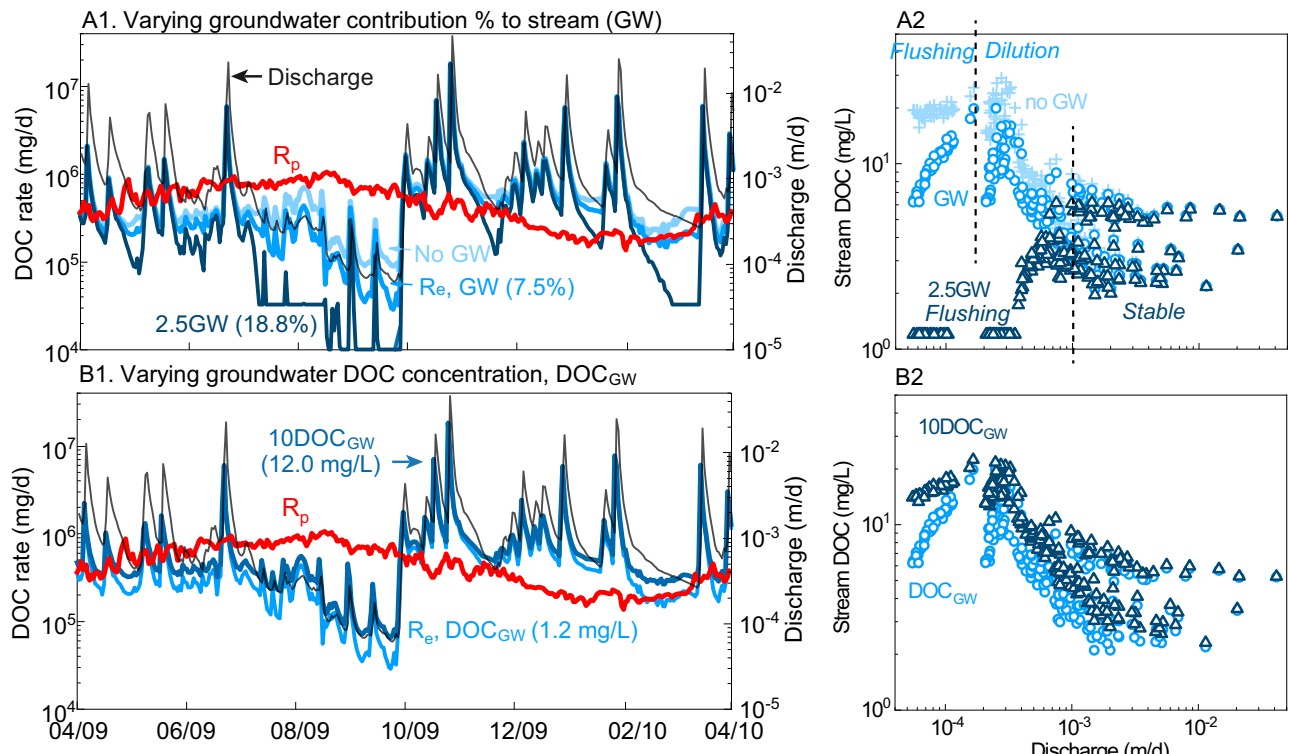

Figure 12. Sensitivity analysis of groundwater on rates ($R_p$ and $R_e$) and C-Q relationships: (A) scenarios with different groundwater volume contribution (%) to stream discharge and (B) scenarios with different groundwater DOC concentration ($DOC_{GW}$). $DOC_{GW}$ and GW ($Q_G/Q$) in the base case was 1.2 mg/L and 7.5%, respectively. 2.5GW in Figure A represents the case with 2.5 times of $Q_G$ compared to the base

case. Increases in the relative groundwater contribution lowered $R_e$ and shifted the C-Q pattern from an overall dilution pattern to an overall flushing pattern; changing $DOC_{GW}$ had negligible influence on DOC rates and C-Q patterns.

Figure 13 summarizes the annual total $R_p$ and $R_e$ in all sensitivity test scenarios. Annual $R_p$ was most
550 sensitive to $T$ compared to $S_w$ and sorption thermodynamics. Annual $R_e$ was less sensitive to $T$ variation though
it also increased with $Q_{10}$ because a higher production led to more export of DOC. Annual $R_p$ also depended on
$f(S_w)$, with the threshold function $f_4(S_w)$ (Section 2.6) having the highest production rates. However, $R_e$ did not
follow the trend of $R_p$ (Figure 13B). Generally, under the same hydrological conditions, a doubling of $R_p$ only led
to about 50% increase in $R_e$. Higher sorption affinity (higher $K_{eq}$) did not change production rates but could reduce
DOC export by about 30% because of large storage of DOC on soils. High relative groundwater inputs (18.8%
versus 7.5%) lowered $R_e$ in all scenarios because more water came from deeper organic-poor groundwater.

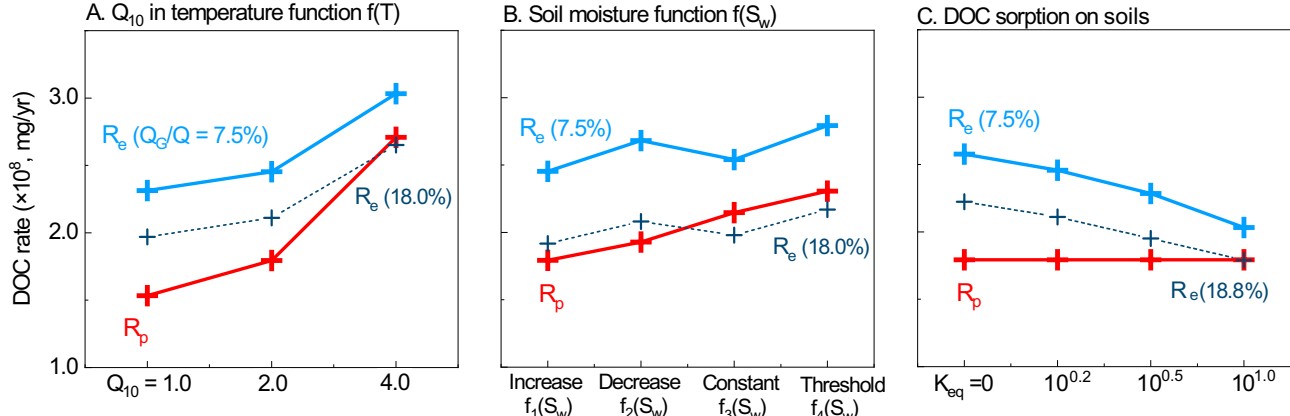

Figure 13. Total $R_p$ (red) and $R_e$ (blue) per year (mg/yr) under two different groundwater volume contribution conditions (7.5% and 18.8%) for three different scenarios: (A) soil $T$, (B) soil moisture, and (C) sorption equilibrium $K_{eq}$. Each symbol represents the annual total from each scenario. The sensitivity of $R_e$ to the reaction characteristics was lower than that of $R_p$. As groundwater flew below the soil-weathered rock interface, changing groundwater contribution does not change $R_p$ such that $R_p$ were identical for the two groundwater contribution cases.

## 4. Discussion

Using field data and a catchment scale reactive transport model, this study revealed that DOC production was primarily regulated by temperature, but the lateral export of DOC was controlled primarily by hydrological conditions. This work contributes to the growing body of research that lateral carbon flux is primarily determined by water routing and hydrological connectivity and only secondarily influenced by biological activity (Zarnetske et al., 2018). Although soil respiration and vertical $CO_2$ fluxes are closely related processes, this work focuses on the net production and export of DOC because it has been studied and understood to a much lesser extent than soil respiration (Tank et al., 2018). To better appreciate the relative importance of land-water-atmosphere carbon fluxes, future research needs to fully integrate lateral DOC fluxes in concert with vertical fluxes of $CO_2$ across terrestrial and freshwater ecosystems.

**DOC production**. Simulated catchment-scale production rate $R_p$ depends more on $T$ than water storage or soil moisture. This finding is expected, as DOC production is biologically mediated and thus, influenced by temperature and the metabolism dependence on temperature (Gillooly et al., 2001). Although the local scale soil moisture varies from 0.40 at the ridge top to 0.70 in swales and riparian zones (Figure 5B), the averaged catchment-scale soil moisture is relatively constant in this temperate humid catchment (0.46 to 0.56, average over the whole catchment), especially compared to places where water is more limited and soil moisture can drop below 0.15 (Korres et al., 2015). This small variation is due to the capability of the shale-derived, clay rich soils

at Shale Hills in holding water and their low dynamic water storage that is responsive to hydroclimatic conditions (Xiao et al., 2019). The influence of soil moisture is likely higher in catchments with more pronounced seasonal changes and more fluctuations in soil moisture.

Our work also suggests that catchment-scale ($R_P$) and local-scale ($r_P$) production rates have different primary controls. The rate law used at the local scale is measured at relatively small spatial scales (0.1-2.0 m in soil pedons) (Bauer et al., 2008; Hongve, 1999; Yan et al., 2016). Our results show that even when the rate law with an optimum soil moisture was used at the local scale ($f_4(S_w)$ in Figure S4B), the catchment-scale rates do not exhibit maximum rates at an "optimal" soil moisture (Figure 8), indicating different controls at the local versus catchment scale. In addition, due to the spatial heterogeneities of $T$, soil moisture and SOC content, the temporal variations of $R_P$ and $r_P$ may be not consistent. At Shale Hills, the daily $R_p$ spanned less than an order of magnitude annually, with its maximum occurring in the dry, hot summer and minimum in the wet, cold winter and spring (Figure 6). Local-scale $r_p$ exhibited similar temporal dynamics but varied by more than 2 orders of magnitude, with rapid production mostly in "hot spots" such as swales and riparian zones with persistently high water content and SOC content (Figure 5). Note that local scale rate laws are often extrapolated directly to catchments or larger scales (Crowther et al., 2016; Conant et al., 2011; Fissore et al., 2009; Moyano et al., 2012). This work suggests that although local scale rate laws have been developed extensively, extrapolation of rates from local to catchment scale may lead to large uncertainties and even errors. This speaks to the importance of understanding controls of biogeochemical transformation rates and developing reaction rate theories at the catchment scale for regional and global scale estimations.

**DOC lateral export.** In contrast to DOC production, this work shows that DOC export is largely driven by hydrological regimes. The DOC concentration measured at the catchment outflow integrates chemical signatures in the connected zones and therefore reflects the temporal and spatial balance of connected zones. The signature of the connected zones depends on catchment size, hydrogeologic structure, vegetation, and climatic setting. Geomorphological and ecological processes have been shown to co-generate systematic differences in the vertical and lateral distribution of SOC and plant biomass, with greater concentration of organic carbon in the valley floor than hillslopes in some catchments (Piney et al., 2018; Temnerud et al., 2016; Campeau et al., 2019; Thomas et al., 2016) and concentrated SOC in uplands in some other catchments (Herndon et al., 2015). These different patterns of SOC distribution may explain the large variation of stream DOC in catchments with different climate conditions (Moatar et al., 2017). The median stream DOC at Shale Hills is relatively high (10.0 mg/L), compared to 3.0 mg/L in temperate humid German catchments (Musolff et al., 2018), 4.1 mg/L in the UK catchments with oceanic climate (Monteith et al., 2015), 4.5 mg/L in France (Moatar et al., 2017), but similar to

9.5 mg/L measured in boreal catchments in Sweden (Winterdahl et al., 2014) and 8.1 mg/L in boreal wet and 10.5 mg/L in boreal dry sites in Norway and Finland (de Wit et al., 2016). These differences suggest that climate, vegetation, and landscape heterogeneity may together shape when, where, and how much the hill is connected to the stream and how much DOC is exported at different times.

**Temporal asynchrony of DOC production and export.** The contrasting temporal pattern of simulated DOC production and export reflects their different control and asynchronous nature of the two processes at the catchment scale. The local DOC production is influenced mostly by the seasonal pattern of soil $T$ whereas export is predominantly controlled by the precipitation events and antecedent conditions, both of which modulate the degree to which DOC production zones are hydrologically connected to the stream. The temporal asynchrony between DOC production and export rates is therefore strongly influenced by the seasonality of temperature and precipitation and local climate. In Shale Hills, the strong weather seasonality amplifies such asynchrony: the wet winter and spring happens to be the cold season whereas the dry summer is hot. In the summer, the catchment essentially produces and stores the DOC in soil water and soil surfaces, and waits for the arrival of the late fall when trees take less water such that large water fluxes routing through the soil and flush out the stored DOC. In other words, low hydrological connectivity in the summer imposes a lag period in DOC export such that the DOC we see today in the stream is often the DOC produced some time ago. In essence, at Shale Hills, the catchment acts as a producer of DOC in summer and a transporter of DOC in late fall and winter when the soil is wetter.

These findings may indicate a strong climatic control over DOC production and export. In places where climate seasonality is not as strong and catchments are hydrologically connected to streams all year long, we may see DOC export all year long and therefore much less extent of asynchrony. In places where we see more seasonality and less hydrological connected places, we may see more asynchrony such that a few high flow events dominate the DOC export. Under Mediterranean climate with strong seasonality, antecedent moisture conditions have been observed to be essential for understanding the temporal pattern of DOC and nutrient (N) export (Bernal et al., 2005, 2002). Hydrological connectivity and water flow paths become dominant as subsurface saturation expands across valley floors and into hillslopes (Covino, 2017; Abbott et al., 2016a). In contrast, our results differ from existing studies at soil pedons showing the synchronization of DOC production and export rates in forest ecosystems of the temperate zone (Michalzik et al., 2001). These differences are likely due to the relatively short water residence time and well-connected systems at the pedon scale.

**DOC storage.** The simulations here suggest that DOC storage depends not only on hydrological connectivity, but also on the sorbing capacity of soils. In this context, clay content and the presence of organo-mineral aggregates might play a role in mediating DOC dynamics (Lehmann et al., 2007; Cincotta et al., 2019).

**Implications for vertical carbon fluxes and other lateral carbon fluxes.** This work focuses on DOC lateral fluxes and does not simulate the carbon loss through soil respiration and associated vertical carbon fluxes of $CO_2$, which has been studied in previous work from the perspective of the carbon budget (Andrews, 2011; Brantley et al., 2018; Hasenmueller et al., 2015). Soil respiration is an important pathway of carbon flux that, similar to DOC production, can be shaped by soil temperature and moisture. Generally, warm temperature and medium soil moisture provide optimal conditions for microbial respiration, leading to significant vertical losses of carbon during summer months (Perdrial et al., 2018; Stielstra et al., 2015). In contrast, low temperature and high soil moisture can hinder aerobic respiration and associated carbon losses as $CO_2$ (Smith et al., 2003), effectively accumulating DOC until large storms flush DOC to streams (Pacific et al., 2010). This pattern is consistent with observations that total $CO_2$ release and DOC production are positively correlated (Neff and Hooper, 2002). The dependence of DOC production and export might also hold true for soil respiration. On the other hand, as part of sorbed DOC may be respired by microbes into $CO_2$, our model might overestimate the DOC accumulation in the catchment, especially in summer.

This work does not consider the transport of particular organic carbon (POC) in soil water and stream water, though POC can play an important role in the carbon budget and biogeochemical cycles in some cases (Ludwig et al., 1996; Diem et al., 2013). In a forested catchment such as SSHCZO, DOC usually comprises the major fraction (between 70-80%) of total organic carbon export (Jordan et al. 1997). The same pattern has been reported in a world review of organic carbon export at the global scale (Alvarez-Cobelas et al., 2012). However, POC export can be significant in human-impacted areas (Correll et al., 2001; Mattsson et al., 2005) and in areas with frequent disturbance (Abbott et al., 2016a). In those cases, it would be important to incorporate POC processes in and consider export of POC that is often strongly influenced by precipitation events and land cover. It may follow a different temporal pattern from DOC, because of difference sources, transportation dynamics, and response to hydrologic regimes (Dhillon and Inamdar, 2014; Alvarez-Cobelas et al., 2012).

**Regulation of C-Q patterns.** During dry periods, stream water mostly reflects the carbon-poor groundwater. As the precipitation wets the catchment, the valley floor area characterized by high SOC and soil water DOC concentration is connected to the stream (Figure 5 and Figure 7). Stream DOC at that time is a mixture of groundwater with low DOC and valley soil water with high DOC. As the catchment becomes wetter and expands the connected zones to the whole valley and swales, the influence of groundwater fades and DOC increases until reaching a threshold connectivity (~0.1 ≈ the valley width / the catchment width). Upon reaching this threshold, a rising contribution of lateral flow from planar hillslopes and uplands characterized by low DOC increased sharply and the stream DOC dropped, leading to a dilution C-Q pattern. In other words, during wet

periods when the whole catchment is hydrologically connected to the stream, stream DOC reflects the "average" concentration across the catchment (~ 2.5 mg/L). The increase and then decrease pattern (or chevron pattern) therefore indicate the presence of three end members from different sources: the groundwater with very low DOC concentration, the soil water in organic-rich swales with highest DOC content, and the uphill soil water with DOC level in between these two.

The overall dilution (or chevron) C-Q pattern observed here with a maximum at a mid-range discharge contrasts the most commonly observed flushing pattern for DOC (Moatar et al., 2017). In fact, it resembles more of the hysteresis behavior often observed in storm and snowmelt events for metals and nutrients (Zhi et al., 2019; Duncan et al., 2017). Previous field studies illustrate that the hydrological connectivity to the stream versus the distribution of SOC ultimately dictates the spatial and temporal dynamics of DOC concentration in soil and stream water, leading to different C-Q relationships (dilution versus flushing) (Bernhardt et al., 2017; Bernal and Sabater, 2012; Covino, 2017). This is illustrated by different C-Q relationships in two head-catchments Shale Hills (US) and Plynlimon (UK) (Herndon et al., 2015). Stream water at Shale Hills is derived from SOC-rich swales with high DOC concentration at low flow and from both swales and hillslopes with low DOC concentration when discharge increases. Conversely, at Plynlimon, SOC is enriched in uplands and therefore concentrations are high at high flow when water flows connect SOC-rich uplands. Our reactive transport modeling provides a quantitative and mechanistic approach to explain the overall dilution behavior of C-Q patterns, which have usually been interpreted as a production/source limitation (Covino, 2017; Zarnetske et al., 2018). Our results are consisted with Covino (2017) proposed mechanisms on driving forces: solute concentrations could increase with connectivity first due to transport limitation while they may decrease with connectivity after a given threshold due to reactivity limitation. Here based on the model, the source limitation essentially means solutes that are enriched in a limited zone within the catchment. Together, these studies suggest that C-Q patterns emerge from a combination of different hydrological and biogeochemical processes, and thus may be much more complicated than initially thought. Modelling approaches such as the one presented here can help us to understand the mechanisms underlying C-Q patterns, and thus improve our ability to predict the evolution of C-Q trajectories under changing climatic conditions.

C-Q patterns also relate to the mixture of different sources of water in the stream, composed of time-varying relative contribution from the shallow soil water and relatively deep groundwater. Their DOC relative contribution can be affected by the vertical distribution of reacting materials (Musolff et al., 2017; Bishop et al., 2004; Seibert et al., 2009; Winterdahl et al., 2016) and the relative volume contribution of source water (soil water vs groundwater below the soil-weathered rock interface) to the stream (Zhi et al., 2019; Radke et al., 2019;

Weigand et al., 2017). With the shale bedrock, the groundwater contribution to the stream is relatively small (~7.5%) at Shale Hills. Soil water (although from a very limited swale area) dominates inflow to the stream even during the summer dry period. When the groundwater volume input increases to about 18.8% of the streamflow by volume (2.5× the actual case), as shown in the sensitivity analysis (Figure 12), the C-Q relationships shift to an overall flushing pattern. This may provide a potential explanation for the DOC C-Q flushing pattern at sandstone-dominant Garner Run (a neighbor catchment of Shale Hills), where the groundwater contributions to the stream are typically higher (Hoagland et al., 2017; Li et al., 2018). More interestingly, this indicates when the groundwater contribution is "sufficiently" high, it might mask the signature of the swale-derived soil water such that the three end-member chevron C-Q pattern become a two end-member pattern with monotonic flushing behavior as observed in Coal Creek where groundwater contributes about 20% annually (Zhi et al., 2019). C-Q relationships have been categorized into 9 patterns, including 3 monotonic and 9 segmented types (Moatar et al., 2017; Underwood et al., 2017). The shifting threshold that separates segments of C-Q responses with the relative groundwater contribution in this work (Figure 12) suggests the relative contribution of groundwater to streamflow may play a pivotal role in shaping the C-Q patterns. This threshold value can potentially provide a rough estimation for the relative contribution of different end-members to the stream.

The mechanisms that regulate DOC C-Q patterns—seasonally variable hydrological connectivity and groundwater contribution—are consistent with previous literature on geogenic species (Mn, Fe), isotopes, and particle fluxes at Shale Hills (Herndon et al., 2018; Kim et al., 2018; Sullivan et al., 2016; Thomas et al., 2013). For example, Mn is associated with DOC via biotic cycling and storage in plant species, and Fe is associated with DOC via aqueous complexation. Both solutes are therefore more abundant in shallow soils. The C-Q pattern of Fe and Mn shows a dilution pattern with concentrations decreasing as discharge increases (Herndon et al., 2015; 2018). In the dry summer, stream water derives from rich-organic swales and riparian zones with high concentrations of soluble Fe and Mn (Herndon et al., 2018), leading to corresponding high stream concentrations. At high flows, these solutes are diluted by the influx of uphill soil water without as much DOC. This again emphasizes the key role of solute sources and hydrological dynamics in controlling stream chemistry.

## 5. Conclusions

The production and export of DOC remain central uncertainties in determining ecosystem-level carbon balance (Raymond et al., 2016; Catalan et al., 2016; Kicklighter et al., 2013). These uncertainties persist despite many studies because there are complex interacting controls on DOC production and export. Indeed, very few studies have quantitatively addressed the linkages between SOC processing, hydrological conditions, and

corresponding DOC processing and export rates at the catchment scale. We found that DOC production was the major DOC source at Shale Hills (76%, compared to 24% from precipitation). Our simulations showed that the temporal dynamics of DOC export rates ($R_e$) were more linked to hydrological flow paths and precipitation events than production rates ($R_p$). Sensitivity analysis further confirmed that $R_p$ was primarily controlled by temperature while $R_e$ was most sensitive to changes in hydrology. This difference in environmental drivers lead to an asynchrony between DOC production and DOC export which was amplified as the summer drought proceeds. During the wet period (spring and winter), the catchment was well connected and DOC production and export occurred simultaneously, while during summer, DOC accumulated in soil pockets disconnected from the stream, and DOC export was limited and constrained to the near stream areas. In other words, the climate seasonality imposes different roles at different time: the catchment serves mostly as a DOC producer in the dry and hot summer but primarily as an exporter in the wet and cold winter.

This work quantitatively demonstrates the key role of hydrological flow paths and the degree of connectivity in determining the C-Q patterns exhibited at the catchment outlet. At low discharges where connectivity is limited (<0.1), stream DOC was mainly sourced from the valley floor and swales which maintained high SOC decomposition rates and soil water DOC concentration. At higher discharges, an increasing relative contribution of soil lateral flow from planar hillslopes and uplands characterized by low soil water DOC, decreasing the stream DOC concentration and therefore exhibiting a dilution C-Q pattern. Although changing the effect of soil $T$, moisture, and sorption on DOC reaction characteristics alters soil water DOC concentration, there is little change in the overall C-Q patterns. However, when groundwater contributes 18.8% of total annual discharge, stream DOC concentration increases with discharge and flushing patterns emerge, emphasizing the significance of relative contribution from different water sources in shaping DOC export patterns. This study provides new insights into how DOC production and export interact at multiple scales, and emphasizes the importance of considering different constraints when projecting the response of lateral and vertical carbon fluxes to climate changes.

**Data availability.** The field data have been digitized and are accessible through national CZO data portal (http://criticalzone.org/shale-hills/data/datasets/). The source code of BFP (BioRT-Flux-PIHM) and the input files necessary to reproduce the results are available from the authors upon request (lili@engr.psu.edu).

**Author contributions.** HW, LL, and all other co-authors conceived the idea and designed the numerical experiments based on ideas generated from a workshop and monthly discussions. HW ran the simulations, analyzed simulation results, and wrote the first draft of the manuscript. All co-authors participated in editing the manuscript.

**Competing interests.** The authors report no conflicts of interest.

**Acknowledgements.** We acknowledge the financial support from the US National Science Foundation Geobiology and Low temperature Geochemistry program via the grant EAR-1724171. We appreciate data from the Susquehanna Shale Hills Critical Zone Observatory (SSHCZO) supported by National Science Foundation Grant EAR – 0725019 (C. Duffy), EAR – 1239285 (S. Brantley), and EAR – 1331726 (S. Brantley). Data were collected in Penn State's Stone Valley Forest, which is funded by the Penn State College of Agriculture Sciences, Department of Ecosystem Science and Management, and managed by the staff of the Forestlands Management Office. We thank the ISU Center for Ecological Research and Education and EPSCoR grant IIA 1301792 that stimulated ideas in this manuscript. SB work was funded by CANTERA (RTI-2018-094521-B-101) and a Ramón y Cajal fellow (RYC-2017-22643) from the Spanish Ministry of Science, Innovation, and Universities.

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
