# Peer review of "Temperature controls production but hydrology controls export of dissolved organic carbon at the catchment scale"

_Hydrology and Earth System Sciences, 2019_

## Referee Comment (RC1) · Anonymous Referee #1 · 12 Aug 2019

Summary

Wen et al. use a well-studied system (the SSHCZO headwater catchment) to model spatial and temporal variation in DOC production and export. The need for this work is outlined in the introduction where the authors argue that production and export are difficult to predict because they are driven by multiple, often competing factors (temperature and hydrology) that complicate outputs. Their work shows that hydrology is the dominant factor influencing DOC export while temperature drives DOC production. The manuscript is well-written and well-detailed and represents a solid contribution to the literature. The authors pull apart complex interactions to model the stream system and provide important insights into the sensitivity of model outputs to variations in different model parameters. I support publication following the below revisions. My main concern to be addressed is that respiration is an important pathway that is not properly considered here.

General comments

Respiration is a potentially major pathway for C loss that is not accounted for in this SSHCZO carbon mass balance model, a fact which is not discussed until the last paragraph of the discussion. Furthermore, the support for not considering respiration is quite weak. No literature on C budgets for SSHCZO is cited, so the "high DOC accumulation" (p.26, l. 44) is not placed in context of other fluxes. That is, there is no quantitative information given here that supports that vertical carbon fluxes are minimal relative to hydrologic export. More consideration for previous work on C budgets at SSHCZO should be cited, starting with the Brantley et al 2018 VZJ review and refs cited therein. If the vertical flux is comparably large, what impact would including this flux have on the observations made in this model?

The conductivity mass balance hydrograph separation used to calculate groundwater input was referred to multiple times but not shown. Is there a supplementary figure that would be useful for supporting this? Please also incorporate more previous literature on surface-groundwater interactions at SSHCZO and how they influence stream chemistry (e.g., Sullivan et al., 2016, Chem Geology; Herndon et al., 2018, Chem Geology; Thomas et al., 2013, VZJ; Kim et at., 2018, EPSL). Do those observations generally match what is observed here for DOC patterns?

This model uses data only from the South Slope – how comparable are these to pore water DOC concentrations on the North Slope? Is it valid to assume that these sites are representative of the entire catchment?

Specific comments

Line numbers are referred to here by page and line number because the full line number was not visible after 100.

p. 4, l. 13. Since Temperature and Precipitation are a large focus of this manuscript, I suggest including well-defined annual T and P values, i.e., the average and standard error for the past ten years.

p. 7, l. 96. Are there citations for these values?

p. 10, l. 65. Should DOC be in mg m$^{-3}$, and was this conversion from L to m$^3$ incorporated?

p.11, l. 7. What percentages do these groundwater inputs correspond to?

p. 12, l. 16. $S_T$ barely changes across the year, so the small $S_T$ during the dry period is not observed. Rather, it looks like it just very slightly decreases relative to wet periods.

p. 12, l. 17. Wouldn't high ET coincide only with shrinking the connected zone, not expanding it?

p. 12, l. 25 (figure caption): Define $S_U$ and $S_S$ in the caption.

p. 13, l. 30. This section states that groundwater contributes substantially to DOC patterns at low discharge. I do not see this in the figure. The stream data and model seem to very closely follow the soil model under all discharge conditions. Is there a way to quantify this contribution and communicate it in the text?

p. 14, l. 64. Does the "legacy of produced DOC…" suggest that DOC is desorbing from the soil in response to flushing?

p. 16, Figure 5. Is DOC mass storage in steady-state over the year?

p. 16, l. 94. The chevron pattern is only observed in the model output, not in the stream data. There are not enough data to support that this pattern occurs at low discharge. I think you can only propose that this pattern would be observed with enough data, but it's not currently supported.

p. 17, l. 00. The explanation for the dilution behavior is clear, but what explains the proposed flushing behavior at low discharge? Swale soil water mixing with groundwater?

p. 19, l. 49. These two lines are contradictory. The first sentence says that sorption "resulted in smaller Re" and the second line says that sorption "increased the magnitude of Re".

p. 20, l. 00. Please clarify…is the increased C storage indefinite? Does high sorption mean that C continues to accumulate in the catchment or is it stored for only a portion of the year and then released (SOC at steady-state)?

p. 22, l. 25. Do these values represent averages over the whole catchment? I would assume there is larger variation in soil moisture between different landscape positions.

p. 23, l. 36. A model like this seems like an interesting way to identify potential hotspots based on temperature and moisture conditions.

Technical comments
l. 101. Suggest "what factors determine"
p. 5, l. 42. Does this mean "Flux-PIHM *separates* the subsurface flow into…"? Awkward as written.
p. 14, l. 58-59. The "Soil water DOC" sentence is not understandable as written.
p. 22, l. 11. Suggest replacing "Rp was identified for…" with "Rp was identical for both groundwater contribution levels…"
p. 24, l. 72. Year for Cincotta ref is missing.

---

## Referee Comment (RC2) · Anonymous Referee #2 · 5 Oct 2019

**Scientific significance**: DOC export is an research important topic that fits well into the scope of HESS. The authors provide an interesting study based on a systematic combination of field data sets and modelling. The model allows to compare the relevance of local process and the catchment-scale effects (e.g., L. 549 – 555) and to evaluate the sensitivity towards different influencing factors. The case study adds to catchment studies such as to expand the understanding how different factors such as climate or local hydrological conditions may influence DOC export.

**Scientific quality:** Overall, the study seems to be carefully done and provides a broad discussion about the relevance and interpretation of the findings. Some aspects are

only presented briefly (see below). This makes it difficult to properly judge all relevant details.

**Presentation quality:** Generally, the paper is well written and easy to understand. However, the method section does only provide incomplete information (see below for more details). Despite having published many methodological aspects before, the manuscript should contain more information to be able to evaluate what the authors have actually done.

**Detailed comments:**

*Abstract:*

From the abstract it remains unclear how well the internal states of the catchment as described by the model are confirmed by field observations. Please explain how well the model performed and what gain in insight was achieved by using the model.

*Introduction:*

L. 55: What is the role of particulate organic matter (POM) in this context? How relevant is it for carbon export and for affecting DOC concentrations?

L. 65: It has been shown in that hyporheic biogeochemical cycles may be more affected by POM than by DOC (Diem *et al.*, 2013).

L. 78 - 79: Why should the conversion to DOC concentration reveal the same pattern? Is this an expectation or confirmed by data analysis?.

L. 93: Can you provide examples for such multiple optima?

L. 99 - 101: What has been done so far to address this issue?

L. 102 - 105: What is the state of the art of DOC modelling by these kind of models? What have others done? What are known limitations? Please provide a short overview that provides context for this work from a modellers perspective.

[Figure]

L. 106 - 113: Why did you select this study area?

L. 109 - 111: To which extent are these expectations based on prior data analyses of measurements in the study area?

*Methods:*

General comment: the methods are described very briefly only. Please provide more information even if the method description has been already published elsewhere.

L. 135: How large were the lysimeters and which depths did they sample?

L. 137: How was DOC measured in the stream? What was the temporal resolution?

L. 190 - 191 : Why is $n$ set to 1.0?

L. 195: Is this exponential decline with depths supported by the data from the catchment? Would one not expect more stepwise changes given the soil profile and horizontation?

L. 201 - 204: Did you consider any temperature-dependence of the thermodynamic equilibrium?

L. 209: How was the model parameterised for the 535 land elements regarding their soil properties?

L. 219: How was the effective macropore conductivity assessed across the entire unsaturated zone?

L. 227: Setting the DOC concentration in groundwater to a fixed value implies that there was no coupling between DOC dynamics in the unsaturated zone and the groundwater in the model?

L. 243: Multi-objective calibration raises a number of questions that haven't been addressed here. Using different variables for joint calibration generally causes the problem of trade-offs between different objective functions leading to Pareto fronts without

one single optimal solution. How did you solve this problem?

L. 245 - 250: Which were all the parameters that were calibrated? What were the ranges of parameter values considered and how was the calibration performed (manually or by any automated procedure)?

L. 278 - 284: According to the text, the ratio $\frac{CV_{DOC}}{CV_Q}$ is always $< 1$. It seems that the categories are only defined based on parameter $b$. Please clarify.

L. 303: Did you assume a constant fraction of groundwater across the entire discharge range? Why did you specifically select 18.8%?

*Results and discussion:*

L: 322: Twice *that of.*

L. 328: Why is high ET coinciding with expanding *AND* shrinking of the connected zone?

L. 349: What does this NSE represent? Is it the average across the NSE values for each of the six sites? Provide these site-specific values as well.

L. 489: What is the meaning of *2.5GW*?

*Figures:*

Fig. 4: The DOC model simulations for the soil DOC values are site-specific. How was this localised model calibration achieved? How was the standard deviation for each data point calculated?

**Recommendation:**

The manuscript provides important and interesting insights and should get published after properly addressing the critical points mentioned above.

**References**

Diem, S., Rudolf von Rohr, M., Hering, J. G., Kohler, H.-P. E., Schirmer, M., von Gunten, U., 2013. NOM degradation during river infiltration: Effects of the climate variables temperature and discharge. Water Research. 47: 6585-6595.

---

## Author Comment (AC1) · 2 Nov 2019

Please see attachment.

Please also note the supplement to this comment:
https://www.hydrol-earth-syst-sci-discuss.net/hess-2019-310/hess-2019-310-AC1-supplement.pdf

---

## Author Comment (AC2) · 2 Nov 2019

**Response to Reviewers' Comments**

We appreciate the efforts of the reviewers for their insightful and constructive comments. We have addressed concerns in the previous round of review. Below, we provide detailed responses to each of the reviewers' comments; and for convenience, we put the reviewer comments in regular font, author responses in blue, and direct quotes from the revised manuscript *in italic*.

**Reviewer #1' comments:**

**Summary**

Wen et al. use a well-studied system (the SSHCZO headwater catchment) to model spatial and temporal variation in DOC production and export. The need for this work is outlined in the introduction where the authors argue that production and export are difficult to predict because they are driven by multiple, often competing factors (temperature and hydrology) that complicate outputs. Their work shows that hydrology is the dominant factor influencing DOC export while temperature drives DOC production. The manuscript is well-written and well-detailed and represents a solid contribution to the literature. The authors pull apart complex interactions to model the stream system and provide important insights into the sensitivity of model outputs to variations in different model parameters. I support publication following the below revisions. My main concern to be addressed is that respiration is an important pathway that is not properly considered here.

**Response:**

We appreciate the reviewer's comments. Please see the point-to-point responses below.

**General comments**

Respiration is a potentially major pathway for C loss that is not accounted for in this SSHCZO carbon mass balance model, a fact which is not discussed until the last paragraph of the discussion. Furthermore, the support for not considering respiration is quite weak. No literature on C budgets for SSHCZO is cited, so the "high DOC accumulation" (p.26, l. 44) is not placed in context of other fluxes. That is, there is no quantitative information given here that supports that vertical carbon fluxes are minimal relative to hydrologic export. More consideration for previous work on C budgets at SSHCZO should be cited, starting with the Brantley et al 2018 VZJ review and refs cited therein. If the vertical flux is comparably large, what impact would including this flux have on the observations made in this model?

**Response:**

We agree that soil respiration is a very important process and the findings from this work have interesting implications for soil respiration. In this work, however, we chose to focus on the net production and export of DOC because we believe these processes are both less understood and less studied than soil respiration. We have now clarified the scope, first in the introduction, Lines 117-119:

"Although soil respiration is an important process, we choose to focus on understanding on the net production and export of DOC in this work."

**and again, in the discussion, Lines 608-612:**

"Although soil respiration and vertical  $CO_2$  fluxes are closely related processes, this work focuses on the net production and export of DOC because it has been studied and understood to a much lesser extent than soil respiration (Tank et al., 2018). To better appreciate the relative importance of land-water-atmosphere carbon fluxes, future research needs to fully integrate lateral DOC fluxes in concert with vertical fluxes of  $CO_2$  across terrestrial and freshwater ecosystems."

We have added citations from previous work on the C budget at SSHCZO, and discussed the C loss through respiration and the potential impact in the first half of the discussion, Lines 682-694:

"Potential implications for vertical carbon fluxes and other lateral carbon fluxes. This work is focused on DOC lateral fluxes, and thus it does not simulate the carbon loss through soil respiration and associated vertical carbon fluxes of CO2 at Shale Hills, which has been studied in previous work from the perspective of the carbon budget (Brantley et al., 2018; Andrews, 2011). Soil microbial respiration is an important pathway of carbon flux that, similar to DOC production, can be shaped by soil temperature and moisture. Generally, warm temperature and medium soil moisture provide optimal conditions for microbial respiration, leading to significant vertical losses of carbon during summer months (Perdrial et al., 2018; Stielstra et al., 2015). In contrast, low temperature and high soil moisture can hinder aerobic respiration and associated carbon losses as CO2 (Smith et al., 2003), effectively accumulating DOC until large storms flush DOC to streams (Pacific et al., 2010). This pattern is consistent with observations that total CO2 release and DOC production are positively correlated (Neff and Hooper, 2002). The dependence of DOC production and export might also hold true for soil respiration. On the other hand, as part of accumulated soil water DOC may be respired by microbes into CO2, our model might overestimate the DOC accumulation in the catchment, especially in summer."

The conductivity mass balance hydrograph separation used to calculate groundwater input was referred to multiple times but not shown. Is there a supplementary figure that would be useful for supporting this? Please also incorporate more previous literature on surface-groundwater interactions at SSHCZO and how they influence stream chemistry (e.g., Sullivan et al., 2016, Chem Geology; Herndon et al., 2018, Chem Geology; Thomas et al., 2013, VZJ; Kim et at., 2018, EPSL). Do those observations generally match what is observed here for DOC patterns?

**Response:**

We have added detail describing the method of conductivity mass balance hydrograph separation and the spatial distribution of soil series at Shale Hills (Figure SI) in the Supporting Information (SI), Lines S32-S36:

"S1. Estimation of groundwater flow  $Q_G$ . Based on estimation in Li et al. (2017), groundwater estimates were refined first by calculating average groundwater fluxes in wet and dry periods using the conductivity mass balance hydrograph separation (Lim et al., 2005) via the online Webbased Hydrograph Analysis Tool (WHAT) (https:// engineerg.purdue.edu/~what). The groundwater influx was further refined by capturing the peaks of stream [DOC], especially under low discharge periods."

Figure. S2. Temporal dynamics of field discharge (dots), groundwater flow  $Q_G$  refined from WHAT (dash line), and corresponding averaged  $Q_G$  in the wet and dry periods (solid line).

We incorporated additional SSHCZO literature and added more discussion on the influence of surface-groundwater interactions on stream chemistry in Lines 760-770:

"The mechanisms that DOC C-Q patterns are regulated by the seasonally variable hydrological connectivity and groundwater contribution are consistent with previous literature on geogenic species (Mn, Fe), isotopes, and particle fluxes at Shale Hills (Herndon et al., 2018; Kim et al., 2018; Sullivan et al., 2016; Thomas et al., 2013). For example, Mn is associated with DOC via biotic cycling and storage in plant species; Fe is associated with DOC via aqueous complexation. Both solutes are therefore more abundant in shallow soils. The C-Q pattern of Fe and Mn shows a chemodynamic (dilution) pattern with concentrations decreasing from low to high discharge conditions (Herndon et al., 2015; Herndon et al., 2018). In the dry summer, stream water derives from rich-organic swales and riparian zones with high concentrations. At high flow regime, they are diluted by the influx of uphill soil water without as much DOC. This emphasizes the key role of solute sources and hydrological dynamics in controlling stream chemistry."

This model uses data only from the South Slope – how comparable are these to pore water DOC concentrations on the North Slope? Is it valid to assume that these sites are representative of the entire catchment?

**Response:**

We don't have soil pore water DOC data on the North Slope in 2008-2010. Due to the heterogeneous catchment characteristics, the sites at the South Slope may not be representative of the entire catchment all the time. We have clarified this in Line 149 and Lines 431-437:

"No soil water DOC samples were collected at the north side of the catchment."

"In August, the average soil T was around  $20^{\circ}$ C. The hydrologically connected zones shrank to the immediate vicinity of the stream, but  $r_p$  increased (about  $2 \times$  from May) at this higher temperature. Simulated soil water [DOC] increased by a factor of 2 across the whole catchment, especially in hillslope and uplands at the north side of the catchment, partly because the produced DOC was trapped in low soil moisture areas that were not hydrologically connected to the stream. This indicates that DOC samples collected at the south side may not be representative of the DOC dynamics over the entire catchment, especially in the summer and fall dry months."

**Specific comments**

Line numbers are referred to here by page and line number because the full line number was not visible after 100.

p. 4, l. 13. Since Temperature and Precipitation are a large focus of this manuscript, I suggest including well-defined annual T and P values, i.e., the average and standard error for the past ten years.

**Response:**

We calculated the annual average T and P values with standard deviations, Lines 139-140:

"The annual average  $\pm$  standard deviation for air T and precipitation is 9.8 $\pm$ 1.9 °C and 1029 $\pm$ 270 mm in the past decade, respectively."

p. 7, l. 76. Are there citations for these values?

**Response:**

We added citations for these values, Lines 212-215:

"k is the kinetic rate constant of net DOC production (=  $10^{-10} \text{ mol/m}^2/\text{s}$ ) (Zhi et al., 2019; Wieder et al., 2014); A is the SOC surface area ( $m^2$ , =  $2.5 \times 10^{-3} \text{ m}^2/\text{g} \times \text{g}$  of SOC mass) which essentially lumps SOC content and biomass abundance (Zhi et al., 2019; Chiou et al., 1990; Kaiser and Guggenberger, 2003)."

p. 10, l. 65. Should DOC be in mg m-3, and was this conversion from L to m3 incorporated?

**Response:**

We modified the units of DOC, Lines 317-318:

"The DOC input from the rainfall  $R_r$  (mg/d) is the precipitation rate (m/d) times the rainfall [DOC] (6.0×10-4 mg/m3 = 0.6 mg/L×10-3 L/m3) and the catchment drainage area (m2)."

p.11, l. 7. What percentages do these groundwater inputs correspond to?

**Response:**

We have added percentages for these groundwater inputs as follows, Lines 367-370:

"Following the conductivity mass balance hydrograph separation (Lim et al., 2005),  $Q_G$  was estimated as  $1.3 \times 10^{-4}$  and  $4.0 \times 10^{-5}$  m/day for the wet and dry periods (August – September), equivalent to 6.9% and 42.2% of the corresponding period-average stream discharge, respectively."

p. 12, l. 16. ST barely changes across the year, so the small ST during the dry period is not observed.

Rather, it looks like it just very slightly decreases relative to wet periods.

**Response:**

It should be Ss, which showed more significant changes in the wet and dry periods, Lines 377-378:

"Generally, the dry period had small Ss, low connectivity and low discharge."

p. 12, l. 17. Wouldn't high ET coincide only with shrinking the connected zone, not expanding it?

**Response:**

We rewrote this sentence, Lines 378-379:

"In other words, high summer ET drove the catchment to drier conditions, therefore decreasing the connectivity to the stream."

p. 12, l. 25 (figure caption): Define SU and SS in the caption.

**Response:**

We defined Su and Ss in the caption, Lines 384-385:

"(*C*) soil water storage  $S_T$  (= unsaturated water storage  $S_u$  + saturated water storage  $S_s$ ) and hydrological connectivity  $I_{cs}$ /Width."

p. 13, 1. 30. This section states that groundwater contributes substantially to DOC patterns at low discharge. I do not see this in the figure. The stream data and model seem to very closely follow the soil model under all discharge conditions. Is there a way to quantify this contribution and communicate it in the text?

**Response:**

We added the temporal dynamics of discharge to Figure 4 for better visualization and quantified the contribution of DOC from groundwater to the stream in the low and high discharge conditions, Lines 393-398:

---

## Author Response (AR1)

**Response to Reviewers' Comments**

We appreciate the efforts of the reviewers for their insightful and constructive comments. We have addressed concerns in the previous round of review. Below, we provide detailed responses to each of the reviewers' comments; and for convenience, we put the reviewer comments in regular font, author responses in blue, and direct quotes from the revised manuscript *in italic*.

**Reviewer #1' comments:**

Summary

Wen et al. use a well-studied system (the SSHCZO headwater catchment) to model spatial and temporal variation in DOC production and export. The need for this work is outlined in the introduction where the authors argue that production and export are difficult to predict because they are driven by multiple, often competing factors (temperature and hydrology) that complicate outputs. Their work shows that hydrology is the dominant factor influencing DOC export while temperature drives DOC production. The manuscript is well-written and well-detailed and represents a solid contribution to the literature. The authors pull apart complex interactions to model the stream system and provide important insights into the sensitivity of model outputs to variations in different model parameters. I support publication following the below revisions. My main concern to be addressed is that respiration is an important pathway that is not properly considered here.

Response:

      We appreciate the reviewer's comments. Please see the point-to-point responses below.

General comments

Respiration is a potentially major pathway for C loss that is not accounted for in this SSHCZO carbon mass balance model, a fact which is not discussed until the last paragraph of the discussion. Furthermore, the support for not considering respiration is quite weak. No literature on C budgets for SSHCZO is cited, so the "high DOC accumulation" (p.26, l. 44) is not placed in context of other fluxes. That is, there is no quantitative information given here that supports that vertical carbon fluxes are minimal relative to hydrologic export. More consideration for previous work on C budgets at SSHCZO should be cited, starting with the Brantley et al 2018 VZJ review and refs cited therein. If the vertical flux is comparably large, what impact would including this flux have on the observations made in this model?

Response:

      We agree that soil respiration is a very important process and the findings from this work have interesting implications for soil respiration. In this work, however, we chose to focus on the net production and export of DOC because we believe these processes are both less understood and less studied than soil respiration. We have now clarified the scope, first in the introduction, Lines 125-126:

*"Although soil respiration is an important process, this study focuses on the net production and export of DOC."*

and again, in the discussion, Lines 571-575:

*"Although soil respiration and vertical $CO_2$ fluxes are closely related processes, this work focuses on the net production and export of DOC because it has been studied and understood to a much lesser extent than soil respiration (Tank et al., 2018). To better appreciate the relative importance of land-water-atmosphere carbon fluxes, future research needs to fully integrate lateral DOC fluxes in concert with vertical fluxes of $CO_2$ across terrestrial and freshwater ecosystems."*

We have added citations from previous work on the C budget at SSHCZO, and discussed the C loss through respiration and the potential impact in the first half of the discussion, Lines 645-657:

***"Implications for vertical carbon fluxes and other lateral carbon fluxes.*** *This work focuses on DOC lateral fluxes and does not simulate the carbon loss through soil respiration and associated vertical carbon fluxes of $CO_2$, which has been studied in previous work from the perspective of the carbon budget (Andrews, 2011; Brantley et al., 2018; Hasenmueller et al., 2015). Soil respiration is an important pathway of carbon flux that, similar to DOC production, can be shaped by soil temperature and moisture. Generally, warm temperature and medium soil moisture provide optimal conditions for microbial respiration, leading to significant vertical losses of carbon during summer months (Perdrial et al., 2018; Stielstra et al., 2015). In contrast, low temperature and high soil moisture can hinder aerobic respiration and associated carbon losses as $CO_2$ (Smith et al., 2003), effectively accumulating DOC until large storms flush DOC to streams (Pacific et al., 2010). This pattern is consistent with observations that total $CO_2$ release and DOC production are positively correlated (Neff and Hooper, 2002). The dependence of DOC production and export might also hold true for soil respiration. On the other hand, as part of sorbed DOC may be respired by microbes into $CO_2$, our model might overestimate the DOC accumulation in the catchment, especially in summer."*

The conductivity mass balance hydrograph separation used to calculate groundwater input was referred to multiple times but not shown. Is there a supplementary figure that would be useful for supporting this? Please also incorporate more previous literature on surface-groundwater interactions at SSHCZO and how they influence stream chemistry (e.g., Sullivan et al., 2016, Chem Geology; Herndon et al., 2018, Chem Geology; Thomas et al., 2013, VZJ; Kim et at., 2018, EPSL). Do those observations generally match what is observed here for DOC patterns?

Response:

We have added detail describing the method of conductivity mass balance hydrograph separation and the spatial distribution of soil series at Shale Hills (Figure S1) in the Supporting Information (SI), Lines S32-S36:

*"**S1. Estimation of groundwater flow $Q_G$.** Based on estimation in Li et al. (2017), groundwater estimates were refined first by calculating average groundwater fluxes in wet and dry periods using the conductivity mass balance hydrograph separation (Lim et al., 2005) via the online Web-based Hydrograph Analysis Tool (WHAT) (https:// engineerg.purdue.edu/~what). The*

*groundwater influx was further refined by capturing the peaks of stream [DOC], especially under low discharge periods."*

[Figure]

*Figure. S2. Temporal dynamics of field discharge (dots), groundwater flow $Q_G$ refined from WHAT (dash line), and corresponding averaged $Q_G$ in the wet and dry periods (solid line).*

We incorporated additional SSHCZO literature and added more discussion on the influence of surface-groundwater interactions on stream chemistry in Lines 722-731:

*"The mechanisms that regulate DOC C-Q patterns—seasonally variable hydrological connectivity and groundwater contribution—are consistent with previous literature on geogenic species (Mn, Fe), isotopes, and particle fluxes at Shale Hills (Herndon et al., 2018; Kim et al., 2018; Sullivan et al., 2016; Thomas et al., 2013). For example, Mn is associated with DOC via biotic cycling and storage in plant species, and Fe is associated with DOC via aqueous complexation. Both solutes are therefore more abundant in shallow soils. The C-Q pattern of Fe and Mn shows a dilution pattern with concentrations decreasing as discharge increases (Herndon et al., 2015; 2018). In the dry summer, stream water derives from rich-organic swales and riparian zones with high concentrations of soluble Fe and Mn (Herndon et al., 2018), leading to corresponding high stream concentrations. At high flows, these solutes are diluted by the influx of uphill soil water without as much DOC. This again emphasizes the key role of solute sources and hydrological dynamics in controlling stream chemistry."*

This model uses data only from the South Slope – how comparable are these to pore water DOC concentrations on the North Slope? Is it valid to assume that these sites are representative of the entire catchment?

Response:

We don't have soil pore water DOC data on the North Slope in 2008-2010. Due to the heterogeneous catchment characteristics, the sites at the South Slope may not be representative of the entire catchment all the time. We have clarified this in Lines 139-140 and Lines 413-419:

*"No soil water DOC samples were collected at the north side of the catchment."*

*"In August, the average soil T increased to around 20°C. The hydrologically-connected zones shrank to the immediate vicinity of the stream, but $r_p$ increased by 2-fold from May. Simulated soil water DOC concentration increased by a factor of 2 across the whole catchment, especially in hillslope and uplands at the north side of the catchment, partly because the produced DOC was trapped in low soil moisture areas that were not hydrologically connected to the stream. This indicates that DOC samples collected at the south side may not well represent the DOC dynamics of the entire catchment, especially in the summer and fall dry months."*

**Specific comments**

Line numbers are referred to here by page and line number because the full line number was not visible after 100.

p. 4, l. 13. Since Temperature and Precipitation are a large focus of this manuscript, I suggest including well-defined annual T and P values, i.e., the average and standard error for the past ten years.

Response:

We calculated the annual average T and P values with standard deviations, Lines 131-132:

*"The annual mean air T is 9.8±1.9 ℃ (±SD) and the annual mean precipitation is 1029±270 mm over the past decade."*

p. 7, l. 76. Are there citations for these values?

Response:

We added citations for these values, Lines 205-208:

*"$k$ is the kinetic rate constant of net DOC production (= $10^{-10}$ mol/m²/s) (Zhi et al., 2019; Wieder et al., 2014); and A is a lumped "surface area" term ($m^2$, = $2.5 \times 10^{-3}$ m²/g × g of SOC mass) that quantifies SOC content and biomass abundance (Chiou et al., 1990; Kaiser and Guggenberger, 2003; Zhi et al., 2019)."*

p. 10, l. 65. Should DOC be in mg m-3, and was this conversion from L to m3 incorporated?

Response:

We modified the units of DOC, Lines 304-306:

*"The DOC input from the rainfall $R_r$ (mg/d) is the precipitation rate (m/d) times the rainfall DOC concentration ($6.0 \times 10^{-4}$ mg/m³ = 0.6 mg/L×$10^{-3}$ L/m³) and the catchment drainage area (m²)."*

p.11, l. 7. What percentages do these groundwater inputs correspond to?

Response:

We have added percentages for these groundwater inputs as follows, Lines 353-355:

*"Following the conductivity mass-balance hydrograph separation (Lim et al., 2005), $Q_G$ was estimated as $1.3 \times 10^{-4}$ and $4.0 \times 10^{-5}$ m/day for the wet and dry periods (August – September), equivalent to 6.9% and 42.2% of average stream discharge in the corresponding times, respectively."*

p. 12, l. 16. ST barely changes across the year, so the small ST during the dry period is not observed.

Rather, it looks like it just very slightly decreases relative to wet periods.

Response:

It should be Ss, which showed more significant changes in the wet and dry periods, Lines 361-362:

*"$S_s$ was negligible in the dry period (close to 0 m), contributing negligibly to the stream."*

p. 12, l. 17. Wouldn't high ET coincide only with shrinking the connected zone, not expanding it?

Response:

We rewrote this sentence, Lines 362-363:

*"High summer ET drove the catchment to drier conditions, therefore decreasing the connectivity to the stream."*

p. 12, l. 25 (figure caption): Define SU and SS in the caption.

Response:

We defined Su and Ss in the caption, Lines 368-369:

*"(C) soil water storage $S_T$ (= unsaturated water storage $S_u$ + saturated water storage $S_s$) and hydrological connectivity $I_{cs}$/Width."*

p. 13, l. 30. This section states that groundwater contributes substantially to DOC patterns at low discharge. I do not see this in the figure. The stream data and model seem to very closely follow the soil model under all discharge conditions. Is there a way to quantify this contribution and communicate it in the text?

Response:

We added the temporal dynamics of discharge to Figure 4 for better visualization and quantified the contribution of DOC from groundwater to the stream in the low and high discharge conditions, Lines 377-382:

[Figure]

*Figure 4. (A) Temporal dynamics of measured and simulated stream [DOC] as well as groundwater and soil water [DOC]. (B)-(G) Local soil water [DOC] for the 6 sampling locations shown in Figure 1B, including 3 planar (panels B-D) and 3 swale locations (panels E-G).*

*"A temporal pattern emerged from changes in the relative contribution of soil water $Q_L$ and groundwater $Q_G$ to stream discharge Q through time. Under dry conditions (e.g., Q < 1.0× 10^{-4} m/day), $Q_G$ contributed substantially to Q (~32-71%; Figure 3), and stream DOC concentration reflected the mixing of groundwater and soil water (Figure 4A), with a contribution from groundwater DOC of 7-17%. Under wet conditions, stream DOC concentration overlapped with soil water DOC concentration (light blue line in Figure 4). Only ~1-8% of stream DOC was sourced from groundwater at these times."*

p. 14, l. 64. Does the "legacy of produced DOC..." suggest that DOC is desorbing from the soil in response to flushing?

Response:

We now clarify this, Lines 421-423:

*"The increase in hydrological connectivity favored the desorption and the flushing of stored DOC, although the soil water DOC concentration remained high because of the large store of sorbed DOC produced during the antecedent dry times."*

p. 16, Figure 5. Is DOC mass storage in steady-state over the year?

Response:

Because the catchment is an open system with temporally changing precipitation and evapotranspiration, the DOC mass storage is not strictly under steady-state conditions

over the year. We have added detail about DOC mass storage over the year in Lines 442-444:

*"The DOC mass storage increased 1.8×10$^6$ mg over the year, about 1.0% of the overall DOC production, which indicated a general mass balance at the catchment scale."*

p. 16, l. 94. The chevron pattern is only observed in the model output, not in the stream data. There are not enough data to support that this pattern occurs at low discharge. I think you can only propose that this pattern would be observed with enough data, but it's not currently supported.

Response:

We have rewritten this, see Lines 453-457:

*"The C-Q relationships showed a slightly positive correlation at low Q followed by a negative correlation at higher Q (Figure 7A). The simulated C-Q relationship captured this trend but overestimated the positive relationship at low Q. The simulated C-Q relationships showed a general dilution behavior with the C-Q slope b = -0.23 and $\frac{CV_{[DOC]}}{CV_Q}$ = 0.22, consistent with the general pattern exhibited in the field data (Figure 7A)."*

p. 17, l. 00. The explanation for the dilution behavior is clear, but what explains the proposed flushing behavior at low discharge? Swale soil water mixing with groundwater?

Response:

We added the following explanation, Lines 461-462:

*"As connectivity and discharge increased and the stream expanded, the contribution of organic-rich swales increased, elevating DOC concentration to its maximum."*

p. 19, l. 49. These two lines are contradictory. The first sentence says that sorption "resulted in smaller Re" and the second line says that sorption "increased the magnitude of Re".

Response:

Thank you. This has been corrected, see Lines 505-507:

*"Simulations showed that strong DOC sorption ($K_{eq}$ = 1.0) did not change $R_p$ but lowered stream DOC concentration and resulted in smaller $R_e$ (Figure 10A). DOC sorption had little impact on $R_p$ but strong sorption decreased the magnitude of $R_e$ by 10%-69%."*

p. 20, l. 00. Please clarify...is the increased C storage indefinite? Does high sorption mean that C continues to accumulate in the catchment or is it stored for only a portion of the year and then released (SOC at steady-state)?

Response:

The increased DOC storage is indefinite, and highly dependent on the precipitation intensity. We have clarified this in Lines 507-510:

> *"The sorbed DOC concentration differed by more than a factor of 3, with more sorbed DOC with larger $K_{eq}$ values (Figure 10B). Large amounts of sorbed DOC persisted until early fall, when large rainfall events flushed out sorbed DOC and reduced DOC storage (Figure 6). This means that catchments can store large quantities of DOC, the specific amount of which depends on sorption capacity."*

p. 22, l. 25. Do these values represent averages over the whole catchment? I would assume there is larger variation in soil moisture between different landscape positions.

**Response:**

These values represent averages over the whole catchment. This is now clarified in Lines 578-584 and Lines 592-596:

> *"Although the local scale soil moisture varies from 0.40 at the ridge top to 0.70 in swales and riparian zones (Figure 5B), the averaged catchment-scale soil moisture is relatively constant in this temperate humid catchment (0.46 to 0.56, average over the whole catchment), especially compared to places where water is more limited and soil moisture can drop below 0.15 (Korres et al., 2015). This small variation is due to the capability of the shale-derived, clay rich soils at Shale Hills in holding water and their low dynamic water storage that is responsive to hydroclimatic conditions (Xiao et al., 2019)."*

> *"At Shale Hills, the daily $R_p$ spanned less than an order of magnitude annually, with its maximum occurring in the dry, hot summer and minimum in the wet, cold winter and spring (Figure 6). Local-scale $r_p$ exhibited similar temporal dynamics but varied by more than 2 orders of magnitude, with rapid production mostly in "hot spots" such as swales and riparian zones with persistently high water content and SOC content (Figure 5)."*

p. 23, l. 36. A model like this seems like an interesting way to identify potential hotspots based on temperature and moisture conditions.

**Response:**

Thank you. We have added this point in Lines 594-601:

> *"Local-scale $r_p$ exhibited similar temporal dynamics but varied by more than 2 orders of magnitude, with rapid production mostly in "hot spots" such as swales and riparian zones with persistently high water content and SOC content (Figure 5). Note that local scale rate laws are often extrapolated directly to catchments or larger scales (Crowther et al., 2016; Conant et al., 2011; Fissore et al., 2009; Moyano et al., 2012). This work suggests that although local scale rate laws have been developed extensively, extrapolation of rates from local to catchment scale may lead to large uncertainties and even errors. This speaks to the importance of understanding controls of biogeochemical transformation rates and developing reaction rate theories at the catchment scale for regional and global scale estimations."*

**Technical comments**

l. 101. Suggest "what factors determine"

**Response:**

We deleted this part so this comment is not relevant any more.

p. 5, l. 42. Does this mean "Flux-PIHM separates the subsurface flow into..."? Awkward as written.

**Response:**

This sentence has been rewritten, Lines 164-165:

*"Flux-PIHM separates the subsurface flow into active interflow in shallow soil zones and groundwater flow deeper than the soil-weathered rock interface."*

p. 14, l. 58-59. The "Soil water DOC" sentence is not understandable as written.

**Response:**

We rewrote this sentence, Lines 415-417:

*"Simulated soil water DOC concentration increased by a factor of 2 across the whole catchment, especially in hillslope and uplands at the north side of the catchment, partly because the produced DOC was trapped in low soil moisture areas that were not hydrologically connected to the stream."*

p. 22, l. 11. Suggest replacing "Rp was identified for..." with "Rp was identical for both groundwater contribution levels..."

**Response:**

This has been replaced, Lines 562-564:

*"As groundwater flew below the soil-weathered rock interface, changing groundwater contribution does not change $R_p$ such that $R_p$ were identical for the two groundwater contribution cases."*

p. 24, l. 72. Year for Cincotta ref is missing.

**Response:**

We corrected it, Lines 643-644:

*"In this context, clay content and the presence of organo-mineral aggregates might play a role in mediating DOC dynamics (Lehmann et al., 2007; Cincotta et al., 2019)."*

**Reviewer #2's comments:**

Anonymous Referee #2

**Scientific significance**: DOC export is a research important topic that fits well into the scope of HESS. The authors provide an interesting study based on a systematic combination of field data sets and modelling. The model allows to compare the relevance of local process and the catchment-scale effects (e.g., L. 549 – 555) and to evaluate the sensitivity towards different influencing factors. The case study adds to catchment studies such as to expand the understanding how different factors such as climate or local hydrological conditions may influence DOC export.

**Scientific quality**: Overall, the study seems to be carefully done and provides a broad discussion about the relevance and interpretation of the findings. Some aspects are only presented briefly (see below). This makes it difficult to properly judge all relevant details

**Presentation quality**: Generally, the paper is well written and easy to understand. However, the method section does only provide incomplete information (see below for more details). Despite having published many methodological aspects before, the manuscript should contain more information to be able to evaluate what the authors have actually done.

Response:
 Thank you for the encouraging comments. Please see the point-to-point responses below.

**Detailed comments:**

*Abstract:*
From the abstract it remains unclear how well the internal states of the catchment as described by the model are confirmed by field observations. Please explain how well the model performed and what gain in insight was achieved by using the model.

Response:
 We added some explanation regarding the model performance in the abstract, Lines 22-26:

 *"Using field measurements of daily stream discharge, evapotranspiration, and stream DOC concentration, we calibrated the catchment-scale biogeochemical reactive transport model BioRT-Flux-PIHM. We used the calibrated model to estimate and compare the daily DOC production rates ($R_p$; the sum of local DOC production rates in individual grid cells) and export rate ($R_e$; the product of concentration and discharge at the stream outlet, or load)."*

 Additionally, see, Lines 45-49:

 *"This study illustrates how different controls of DOC production and export - temperature and hydrological flow path, respectively - can create temporal asynchrony at the catchment scale. Future warming and increasing hydrological extremes could accentuate this asynchrony, with*

*DOC production occurring primarily during dry periods and lateral export of DOC dominated by a few major storm events."*

*Introduction:*
L. 55: What is the role of particulate organic matter (POM) in this context? How relevant is it for carbon export and for affecting DOC concentrations?

L. 65: It has been shown in that hyporheic biogeochemical cycles may be more affected by POM than by DOC (Diem et al., 2013).

**Response to the above two comments:**
> **We have added some discussion on particulate organic carbon (POC) in Lines 658-667:**

> *"This work does not consider the transport of particular organic carbon (POC) in soil water and stream water, though POC can play an important role in the carbon budget and biogeochemical cycles in some cases (Ludwig et al., 1996; Diem et al., 2013). In a forested catchment such as SSHCZO, DOC usually comprises the major fraction (between 70-80%) of total organic carbon export (Jordan et al. 1997). The same pattern has been reported in a world review of organic carbon export at the global scale (Alvarez-Cobelas et al., 2012). However, POC export can be significant in human-impacted areas (Correll et al., 2001; Mattsson et al., 2005) and in areas with frequent disturbance (Abbott et al., 2016a). In those cases, it would be important to incorporate POC processes in and consider export of POC that is often strongly influenced by precipitation events and land cover. It may follow a different temporal pattern from DOC, because of difference sources, transportation dynamics, and response to hydrologic regimes (Dhillon and Inamdar, 2014; Alvarez-Cobelas et al., 2012)."*

L. 78 - 79: Why should the conversion to DOC concentration reveal the same pattern? Is this an expectation or confirmed by data analysis?

**Response:**
> **To avoid confusion, this has been removed.**

L. 93: Can you provide examples for such multiple optima?

**Response:**
> **We provided examples in Lines 86-89:**

> *"DOC production rates can exhibit low temperature sensitivity in highly weathered soils with high clay content (Davidson and Janssens, 2006). They have also shown to increase with soil water content in sandy-loam soils (Yuste et al., 2007) and to have an optimum with volumetric water content ~0.75 in fine sands (Skopp et al., 1990)."*

L. 99 - 101: What has been done so far to address this issue?

L. 102 - 105: What is the state of the art of DOC modelling by these kind of models? What have others

done? What are known limitations? Please provide a short overview that provides context for this work from a modellers perspective.

**Response to the above two comments:**
    More details have been added, see Lines 95-112:

*"One flexible approach to understanding DOC production and export is the use of reactive transport modeling (RTM). These models integrate multiple production, consumption, and export processes, potentially allowing quantification of individual and coupled processes (Steefel et al., 2015; Li, 2019). The use of RTMs complements statistical regression tools for identification of factors influencing DOC dynamics (Correll et al., 2001; Herndon et al., 2015; Zarnetske et al., 2018). Historically, RTMs have been used in groundwater systems, where direct observations are particularly challenging (Kolbe et al., 2019; Wen and Li, 2018). At the catchment scale, biogeochemical modules have been developed as add-ons to hydrological models. For example, a DOC production module was coupled to the HBV hydrological model, using a static SOC pool that emphasized the influence of active-layer dynamics and slope aspect (Lessels et al., 2015). The INCA-C (Futter et al., 2007) and extended LPJ-GUESS (Tang et al., 2018) models have investigated the importance of land cover in determining DOC terrestrial routing and lateral transport. Terrestrial and aquatic carbon processes have also been integrated into the Soil and Water Assessment Tool (SWAT) to simulate aquatic DOC dynamics (Du et al., 2019). These modules typically simulate individual reactions without considering multi-elemental thermodynamics and kinetics.*
    *In this context, the recently-developed BFP model (Biogeochemical Reactive Transport - Flux - Penn State Integrated Hydrologic Modeling System, BioRT-Flux-PIHM) fills an important gap by incorporating coupled elemental cycling, stoichiometry, and rigorous thermodynamics and kinetics (Bao et al., 2017; Zhi et al., 2019)."*

L. 106 - 113: Why did you select this study area?

**Response:**
    The reasoning has been explained in Lines 113-122:

*"We applied the BFP to a temperate forest catchment in the Susquehanna Shale Hills Critical Zone Observatory (SSHCZO) with extensive data. This small catchment (<0.1 km$^2$) has gentle topography with a network of shallow depressions or swales that have high SOC and deep soils (detailed in Section 2). It is underlain with only with one type of lithology (shale) and land use (forest), providing a useful testbed to evaluate biogeochemical and hydrological functions (Brantley et al., 2018). Previous lab and field work have identified non-chemostatic C-Q patterns of DOC at SSHCZO that are attributable to differences in the hydrologic connectivity of organic-rich soils during different flow conditions (Andrews et al., 2011; Herndon et al., 2015). SSHCZO has spatially-extensive and high-frequency measurements of soil properties, hydrology, and biogeochemistry (Brantley et al., 2018). These data facilitate detailed benchmarking of the BFP model and evaluation of processes controlling DOC production and export."*

L. 109 - 111: To which extent are these expectations based on prior data analyses of measurements in the study area?

**Response:**

We added more descriptions regarding the findings from previous lab and field work in Lines 117-120:

*"Previous lab and field work have identified non-chemostatic C-Q patterns of DOC at SSHCZO that are attributable to differences in the hydrologic connectivity of organic-rich soils during different flow conditions (Andrews et al., 2011; Herndon et al., 2015)."*

Methods:

General comment: the methods are described very briefly only. Please provide more information even if the method description has been already published elsewhere.

Response:

We have added modeling details to the methods section. See responses below.

L. 135: How large were the lysimeters and which depths did they sample?

Response:

We added detail in Lines 135-139:

*"We collected soil water DOC samples with lysimeters with a diameter of 5 cm installed at 10- or 20-cm intervals from the soil surface to a depth of hand-auger refusal, which varied from 30 to 160 cm depending on soil thickness. There were a total of six sampling locations (Figure 1B), including three at the south planar sites—valley floor (SPVF), midslope (SPMS), and ridgetop (SPRT)—and three at the swale sites—valley floor (SSVF), midslope (SSMS), and ridgetop (SSRT)."*

L. 137: How was DOC measured in the stream? What was the temporal resolution?

Response:

We added more description to Lines 140-142:

*"Stream water DOC samples were collected daily in grass bottles at the weir of the stream outlet. All soil water and stream water DOC samples were then filtered (0.45 µm Nylon syringe filters) and were analyzed with a Shimadzu TOC-5000A analyzer (detailed in Andrews et al. (2011))."*

L. 190 -191: Why is n set to 1.0?

Response:

We now clarify this in Lines 212-214:

*"The $f(S_w)$ has the form $f(S_w) = (S_w)^n$ in the base case, where n is the saturation exponent with a value of 1.0, which is within the typical range of 0.75-3.0 for most soils (Yan et al., 2018; Hamamoto et al., 2010)."*

L. 195: Is this exponential decline with depths supported by the data from the catchment? Would one not expect more stepwise changes given the soil profile and horizontation?

Response:

This exponential decline with depth is supported by the data. And we agree with the reviewer that this pattern of decline may vary under different natural conditions. This has been clarified, see Lines 217-222:

*"SOC content typically decreases with depth (Billings, 2018; Bishop et al., 2004), though the specific pattern may vary with soil texture, landscape position, vegetation, and climate (Jobbagy and Jackson, 2000). The depth function of SOC at Shale Hills has been observed to be exponential (Andrews et al., 2011), which is typical of many soils (Billings et al., 2018; Currie et al., 1996). To take this into account, we use the equation $C_d(z) = C_0 exp\left(-\frac{z}{b_m}\right)$, where $C_d$ is SOC at depth z below the surface; $C_0$ is the SOC level at the ground surface and $b_m$ quantified the decline rate with depth, set here to a value of 0.3 (Weiler and McDonnell, 2006)."*

L. 201 - 204: Did you consider any temperature-dependence of the thermodynamic equilibrium?

Response:

Considering the lack of data and previous experimental work showing the low dependence of DOC sorption on temperature (Kaiser et al., 2001), we do not consider the $K_{eq}$ change with temperature. This has been clarified in Lines 228-233:

*"The $K_{eq}$ value represents the thermodynamic limit of the sorption, i.e., the sorption affinity of the soil for DOC. It depends on temperature but also soil properties such as the content of clay and iron oxides (Kaiser et al., 2001; Conant et al., 2011). A $K_{eq}$ value of $10^{0.2}$ was obtained by fitting the stream and soil water DOC data (detailed in Section 2.4).The sum of $[\equiv X]$ and $[\equiv XDOC]$ represents the sorption capacity of the soil with a value ranging from $4.0 \times 10^{-5}$ - $6.0 \times 10^{-5}$ mol/g soil at Shale Hills (Jin et al., 2010; Li et al., 2017), depending on the mineralogy."*

L. 209: How was the model parameterized for the 535 land elements regarding their soil properties?

Response:

We have added detail to the methods section (Lines 244-251), and Figure S1 and Table S1 in the SI:

*"In addition, extensive characterization and measurement data at Shale Hills have been used to define soil depth and soil mineralogical properties such as surface area, and ion exchange capacity that are heterogeneously distributed across the catchment (Andrews et al., 2011; Lin, 2006; Jin and Brantley, 2011; Jin et al., 2010; Shi et al., 2013) (criticalzone.org/shale-hills/data/). Other soil matrix properties include conductivity, porosity, and van Genuchten parameters. Soil macropores such as cracks, fractures, and roots can generate preferential flows. Their properties are represented using the area macropore fraction, depth, and conductivities. They are parameterized based on values quantified in previous studies at Shale Hills (Shi et al., 2013; Lin, 2006), shown in Figure S1 and Table S1."*

Figure S1 in the SI shows the spatial distribution of soil series:

[Figure]

Ernest
Blairton
Rushtown
Berks
Weikert

*Figure S1. Spatial distribution of soil series at Shale Hills.*

Table S1 in the SI lists the parameters for soil properties in the model:

*Table S1. Soil parameters. Listed values are the a priori (uncalibrated) parameter values. All parameters in this table are calibrated using an optimization algorithm.*

| Parameter | Description | Soil type | | | | | Source |
|---|---|---|---|---|---|---|---|
| | | Weikert | Berks | Rushtown | Blairton | Ernest | |
| $K_{infV}$ | Vertical saturated hydraulic conductivity of infiltration layer (m/s) | 9.1 | 15.2 | 9.8 | 1.5 | 8.3 | (Lin, 2006) |
| $K_V$ | Vertical saturated hydraulic conductivity (m/s) | 1.6 | 1.9 | 1.1 | 0.7 | 3.7 | (Lin, 2006) |
| $K_H$ | Horizontal saturated hydraulic conductivity (m/s) | 1.2 | 1.0 | 2.3 | 3.0 | 7.0 | (Lin, 2006) |
| $\phi$ | Porosity $(m^3/m^3)$ | 0.37 | 0.40 | 0.42 | 0.41 | 0.49 | (Lin, 2006) |
| $\phi_r$ | Residual porosity $(m^3/m^3)$ | 0.05 | 0.05 | 0.05 | 0.05 | 0.05 | (Lin, 2006) |
| $\alpha$ | Van Genuchten soil parameter $(m^{-1})$ | 8.80 | 6.45 | 6.50 | 5.34 | 5.82 | (Lin, 2006) |
| $\beta$ | Van Genuchten soil parameter (-) | 1.24 | 1.21 | 1.26 | 1.26 | 1.22 | (Lin, 2006) |
| $f_{mac,V}$ and $f_{mac,H}$ | Vertical and horizontal area fraction of macropores $(m^2/m^2)$ | 0.01 | | | | | Empirical (Shi et al., 2013) |
| $D_{mac}$ | Macropore depth (m) | 1.0 | | | | | Empirical (Shi et al., 2013) |
| $K_{mac,V}$ | Vertical macropore hydraulic conductivity $(m/s)^a$ | 100 $K_{infV}$ | | | | | Empirical (Shi et al., 2013) |
| $K_{mac,H}$ | Horizontal macropore hydraulic conductivity $(m/s)^a$ | 1000 $K_H$ | | | | | Empirical (Shi et al., 2013) |

*a. Soil horizontal macropore hydraulic conductivity and soil vertical macropore hydraulic conductivity are assumed to be 1000 and 100 times their corresponding soil matrix conductivities, respectively.*

L. 219: How was the effective macropore conductivity assessed across the entire unsaturated zone?

Response:

The macropore conductivity was assessed based on the corresponding soil matrix conductivity. See added detail in the methods section (Lines 248-251), and Figure S1 and Table S1 (shown in the previous response):

*"Soil macropores such as cracks, fractures, and roots can generate preferential flows. Their properties are represented using the area macropore fraction, depth, and conductivities. They are*

*parameterized based on values quantified in previous studies at Shale Hills (Shi et al., 2013; Lin, 2006), shown in Figure S1 and Table S1."*

L. 227: Setting the DOC concentration in groundwater to a fixed value implies that there was no coupling between DOC dynamics in the unsaturated zone and the groundwater in the model?

**Response:**

It is true that for this version of the model, the groundwater is a separate input to the stream and is decoupled. This has been clarified, see Lines 170-173 and Lines 189-191:

*"In this version of Flux-PIHM, the deeper groundwater flow $Q_G$ is a separate input to the stream and is decoupled from the shallow soil water. This is supported by field data that shows negligible seasonal variation in groundwater chemistry (Jin et al., 2014; Thomas et al., 2013; Kim et al., 2018)."*

*"In the vertical direction, soil pores are not saturated with water in the shallow unsaturated zone and water flows vertically until it reaches the saturated zone where water forms interflows and moves laterally to the stream."*

L. 243: Multi-objective calibration raises a number of questions that haven't been addressed here. Using different variables for joint calibration generally causes the problem of trade-offs between different objective functions leading to Pareto fronts without one single optimal solution. How did you solve this problem?

L. 245 - 250: Which were all the parameters that were calibrated? What were the ranges of parameter values considered and how was the calibration performed (manually or by any automated procedure)?

**Response to the above two comments:**

The previous hydrological modeling work at Shale Hills (Shi et al., 2013; Li et al., 2017) captured the temporal dynamics of stream discharge well (NSE>0.6). In this work, the main purpose for refining groundwater flow in the wet and dry periods, based on the estimation from Li et al., (2017) and hydrography separation, is to reproduce the stream DOC data. We have not performed multi-objective calibration here. This is now clarified in Lines 271-283:

*"To reproduce the DOC data, we first set the SOC surface area A using a literature range of $10^{-3}$-$10^0$ $m^2$/g (Zhi et al., 2019; Chiou et al., 1990; Kaiser and Guggenberger, 2003). We also set $K_{eq}$ using a literature range of $10^0$-$10^1$ (Oren and Chefetz, 2012; Ling et al., 2006). Once the simulated output captured the temporal trend of data, we refined $Q_G$ based on the estimation from hydrograph separation (Figure S2) to capture the peaks of stream DOC concentration, especially under low discharge periods. Because not all soils are in contact with water, the calibrated surface area represents the effective solid-water contact area in a heterogeneous subsurface, and is orders of magnitude lower than the reported SOC surface areas from laboratory experiments (Kaiser and Guggenberger, 2003). The calibrated hydrological parameters are mostly from Shi et al. (2013), except groundwater estimation. With the overall groundwater flow estimated in Li et al. (2017), groundwater estimates were further refined by calculating average groundwater fluxes in wet and dry periods using conductivity mass-balance hydrograph separation (Lim et al., 2005) and then by reproducing the stream DOC concentration. In other words, stream and*

L. 278 - 284: According to the text, the ratio $CV_{DOC}$ is always < 1. It seems that the CVQ categories are only defined based on parameter *b*. Please clarify.

Response:

This is now clarified in Lines 308-318:

*"C-Q patterns were quantified using two complementary approaches: the power law equation $C = aQ^b$ (Godsey et al., 2009) and the ratio of coefficient of variations of DOC concentration and discharge $\frac{CV_{[DOC]}}{CV_Q}$ (Musolff et al., 2015). We used both methods because the slope of the power law equation does not account for the goodness-of-fit of the C-Q pattern itself. For example, a slope of b = 0 would be considered chemostatic (i.e. relatively small variation of concentration compared to discharge), although high variability in the solute concentration would actually render the behavior chemodynamic (i.e., solute concentrations are sensitive to changes in discharge) (Musolff et al., 2015). We considered two general categories based on these metrics (Godsey et al., 2009; Underwood et al., 2017; Musolff et al., 2015): If b fall between -0.2 and 0.2 and $\frac{CV_{[DOC]}}{CV_Q} \ll 1$, C-Q patterns were considered chemostatic; Values of $|b| > 0.2$ or $\frac{CV_{[DOC]}}{CV_Q} \geq 1$, indicated a chemodynamic behavior. In the chemodynamic category, values of b>0.2 indicate flushing, while values of b < -0.2 indicate dilution."*

L. 303: Did you assume a constant fraction of groundwater across the entire discharge range? Why did you specifically select 18.8%?

Response:

The fraction of groundwater across the entire range of discharge is not constant. In the sensitivity analysis, we first assigned a constant multiplier of groundwater flow rate (0× and 2.5×) based on the base case, rather than a constant fraction of daily groundwater over daily discharge. The 18.8% is corresponding to the case with the groundwater flow rate of 2.5×. It represents the overall fraction of groundwater to annual discharge. We rewrote this sentence for clarification in Lines 335-338:

*"The groundwater flow rates were varied from negligible ($Q_G$ = 0) to 2.5 times of those at the base case ($Q_G$ = 3.3×10⁻⁴ and 1.0×10⁻⁴ m/day the wet and dry periods, respectively). The corresponding fractions ($Q_G/Q$) of groundwater flow to the total annual discharge for the two cases were 0 and 18.8%, respectively."*

*Results and discussion:*
L: 322: Twice *that of*.

Response:

We corrected it, Lines 357-358:

*"In the dry months from August to September, the stream was almost dry with no visible flow and the relative contribution of groundwater to discharge was comparable to that of $Q_L$ (Figure 3B)."*

L. 328: Why is high ET coinciding with expanding AND shrinking of the connected zone?

Response:

We rewrote it, Lines 362-363:

*"High summer ET drove the catchment to drier conditions, therefore decreasing the connectivity to the stream."*

L. 349: What does this NSE represent? Is it the average across the NSE values for each of the six sites? Provide these site-specific values as well.
`

Response:

We rewrote this part and provided specific NSE values for each site, Lines 385-387:

*"The simulated soil water DOC at local scales captured this less-variation trend and the overall model performance was acceptable (i.e., NSE >0.5), though the goodness-of-fit was lower for some locations (e.g. NSE value of 0.36 (SPRT), 0.42 (SPMS), 0.60 (SPVF), 0.46 (SSRT), 0.40 (SSMS), and 0.51 (SSVF))"*

L. 489: What is the meaning of 2.5GW?

Response:

We have clarified this in the caption of Figure 12, Lines 544-545:

*"2.5GW in Figure A represents the case with 2.5 times of $Q_G$ compared to the base case."*

*Figures:*
Fig. 4: The DOC model simulations for the soil DOC values are site-specific. How was this localized model calibration achieved? How was the standard deviation for each data point calculated?

Response:

We tuned the SOC surface area $A$ and the equilibrium constant $K_{eq}$ of DOC sorption to reproduce soil water [DOC] across all sites rather than specifically for each individual local site. The model outputs at local scales did not all achieve the acceptable performance (i.e., NSE>0.5) compared to the corresponding field measurements, due to the uncaptured local heterogeneities. This is clarified and the possible reasons for the discrepancy between the model and measurements are discussed on Lines 271-276 and Lines 385-394:

*"To reproduce the DOC data, we first set the SOC surface area $A$ using a literature range of $10^{-3}$-$10^{0}$ $m^2$/g (Zhi et al., 2019; Chiou et al., 1990; Kaiser and Guggenberger, 2003). We also set $K_{eq}$*

*using a literature range of $10^0$-$10^1$ (Oren and Chefetz, 2012; Ling et al., 2006). Once the simulated output captured the temporal trend of data, we refined $Q_G$ based on the estimation from hydrograph separation (Figure S2) to capture the peaks of stream DOC concentration, especially under low discharge periods."*

*"The simulated soil water DOC at local scales captured this less-variation trend and the overall model performance was acceptable (i.e., NSE >0.5), though the goodness-of-fit was lower for some locations (e.g. NSE value of 0.36 (SPRT), 0.42 (SPMS), 0.60 (SPVF), 0.46 (SSRT), 0.40 (SSMS), and 0.51 (SSVF)). This discrepancy between overall and partial model performance may be due to local variation in soil properties and organic carbon content for which we do not have detailed information. Although the model explicitly considered spatial heterogeneities such as topography and soil properties, averaged values represented grid sizes from 10 to 100 m, and this local scale was large compared to field sampling size (e.g., lysimeters with a diameter of 5 cm). Geochemical processes are sensitive to local properties, including SOC%, SOC surface area and sorption sites, while the representation of these properties was based on a few measurements that were only coarsely defined as ridgetop, midslope, and valley floor."*

We added more details on the calculation the standard deviation in the caption of Figure 4, Lines 403-405:

*"The mean ± standard deviation for each location was calculated based on samples taken at different depths with 10- or 20-cm intervals from the soil surface down to depth of hand-augering refusal."*

**Recommendation:**
The manuscript provides important and interesting insights and should get published after properly addressing the critical points mentioned above.

**References**

[revised manuscript text omitted]

---

## Author Response (AR2)

**Response to Editor' Comments**

We appreciate the efforts of the editor Dr. Christian Stamm for handling and reviewing the manuscript. We have addressed concerns in the previous round of review. Below, we provide detailed responses to the editor's comments; and for convenience, we put the editor's comments in regular font, author responses in blue, and direct quotes from the revised manuscript *in italic*.

**Editor's comments:**

Thank you submitting the revised version of this manuscript. I have to admit that it was quite tedious to identify the actual changes in the revised version because your response to the reviews (including the suggested changes and the respective lines numbers) and the actual changes do not correspond well to each other.

Nevertheless, you have addressed issues that were raised with one exception. Rev.2 asked for a statement on model performance in the abstract. You proposed a modified text ("The model was calibrated using field measurements of daily stream discharge, evapotranspiration, and stream DOC concentrations and met the satisfactory standard of the Nash-Sutcliffe efficiency (NSE) > 0.5. The calibrated model was used to estimate and compare the daily DOC production rates (Rp; the sum of local DOC production rates in individual grid cells) and the daily DOC export rates (Re; the product of concentration and discharge at the stream outlet, or load)."): However, the revised version lacks the information on model performance. Please include this into the revised version.

**Response:**

We apologize for the inconvenience. We included the information on model performance in the abstract, Lines 22-26:

"Using field measurements of daily stream discharge, evapotranspiration, and stream DOC concentration, we calibrated the catchment-scale biogeochemical reactive transport model BioRT-Flux-PIHM, which met the satisfactory standard of the Nash-Sutcliffe efficiency (NSE) > 0.5. We used the calibrated model to estimate and compare the daily DOC production rates ( $R_p$ ; the sum of local DOC production rates in individual grid cells) and export rate ( $R_e$ ; the product of concentration and discharge at the stream outlet, or load)".

**Temperature controls production but hydrology controls export of dissolved organic carbon at the catchment scale**

Hang Wen1, Julia Perdrial2, Benjamin W. Abbott3, Susana Bernal4, Rémi Dupas5, Sarah E. Godsey6, Adrian Harpold7, Donna Rizzo8, Kristen Underwood8, Thomas Adler2, Gary Sterle7, Li Li1\*

[revised manuscript text omitted]

365 the connectivity to the stream.